# Dual-Free Stochastic Decentralized Optimization with Variance Reduction

**Hadrien Hendrikx**
INRIA - DIENS - PSL Research University
hadrien.hendrikx@inria.fr

**Francis Bach**
INRIA - DIENS - PSL Research University
francis.bach@inria.fr

**Laurent Massoulié**
INRIA - DIENS - PSL Research University
laurent.massoulie@inria.fr

## Abstract

We consider the problem of training machine learning models on distributed data in a decentralized way. For finite-sum problems, fast single-machine algorithms for large datasets rely on stochastic updates combined with variance reduction. Yet, existing decentralized stochastic algorithms either do not obtain the full speedup allowed by stochastic updates, or require oracles that are more expensive than regular gradients. In this work, we introduce a Decentralized stochastic algorithm with Variance Reduction called DVR. DVR only requires computing stochastic gradients of the local functions, and is computationally as fast as a standard stochastic variance-reduced algorithms run on a $1/n$ fraction of the dataset, where $n$ is the number of nodes. To derive DVR, we use Bregman coordinate descent on a well-chosen dual problem, and obtain a dual-free algorithm using a specific Bregman divergence. We give an accelerated version of DVR based on the Catalyst framework, and illustrate its effectiveness with simulations on real data.

## 1 Introduction

We consider the regularized empirical risk minimization problem distributed on a network of $n$ nodes. Each node has a local dataset of size $m$, and the problem thus writes:

$$\min_{x \in \mathbb{R}^d} F(x) \triangleq \sum_{i=1}^n f_i(x), \text{ with } f_i(x) \triangleq \frac{\sigma_i}{2}\|x\|^2 + \sum_{j=1}^m f_{ij}(x), \tag{1}$$

where $f_{ij}$ typically corresponds to the loss function for training example $j$ of machine $i$, and $\sigma_i$ is the local regularization parameter for node $i$. We assume that each function $f_{ij}$ is convex and $L_{ij}$-smooth (see, *e.g.*, [30]), and that each function $f_i$ is $M_i$-smooth. Following [40], we denote $\kappa_i = (1 + \sum_{i=1}^m L_{ij})/\sigma_i$ the stochastic condition number of $f_i$, and $\kappa_s = \max_i \kappa_i$. Similarly, the batch condition number is $\kappa_b = \max_i M_i/\sigma_i$. It always holds that $\kappa_b \leq \kappa_s \leq m\kappa_b$, but generally $\kappa_s \ll m\kappa_b$, which explains the success of stochastic methods. Indeed, $\kappa_s \approx m\kappa_b$ when all Hessians are orthogonal to one another which is rarely the case in practice, especially for a large dataset.

Regarding the distributed aspect, we follow the standard *gossip* framework [5, 29, 8, 32] and assume that nodes are linked by a communication network which we represent as an undirected graph $G$. We denote $\mathcal{N}(i)$ the set of neighbors of node $i$ and $\mathbb{1} \in \mathbb{R}^d$ the vector with all coordinates equal to 1. Communication is abstracted by multiplication by a positive semi-definite matrix $W \in \mathbb{R}^{n \times n}$, which is such that $W_{k\ell} = 0$ if $k \notin \mathcal{N}(\ell)$, and $\text{Ker}(W) = \text{Span}(\mathbb{1})$. The matrix $W$ is called the *gossip matrix*, and we denote its spectral gap by $\gamma = \lambda_{\min}^+(W)/\lambda_{\max}(W)$, the ratio between the smallest

non-zero and the highest eigenvalue of $W$, which is a key quantity in decentralized optimization. We finally assume that nodes can compute a local stochastic gradient $\nabla f_{ij}$ in time 1, and that communication (*i.e.*, multiplication by $W$) takes time $\tau$.

**Single-machine stochastic methods.** Problem (1) is generally solved using first-order methods. When $m$ is large, computing $\nabla F$ becomes very expensive, and batch methods require $O(\kappa_b \log(\varepsilon^{-1}))$ iterations, which takes time $O(m\kappa_b \log(\varepsilon^{-1}))$, to minimize $F$ up to precision $\varepsilon$. In this case, updates using the stochastic gradients $\nabla f_{ij}$, where $(i,j)$ is selected randomly, can be much more effective [4]. Yet, these updates are noisy and plain stochastic gradient descent (SGD) does not converge to the exact solution unless the step-size goes to zero, which slows down the algorithm. One way to fix this problem is to use variance-reduced methods such as SAG [33], SDCA [35], SVRG [16] or SAGA [7]. These methods require $O((nm + \kappa_s) \log(\varepsilon^{-1}))$ stochastic gradient evaluations, which can be much smaller than $O(m\kappa_b \log(\varepsilon^{-1}))$.

**Decentralized methods.** Decentralized adaptations of gradient descent in the smooth and strongly convex setting include EXTRA [37], DIGing [28] or NIDS [21]. These algorithms have sparked a lot of interest, and the latest convergence results [14, 42, 20] show that EXTRA and NIDS require time $O((\kappa_b + \gamma^{-1})(m + \tau)) \log(\varepsilon^{-1}))$ to reach precision $\varepsilon$. A generic acceleration of EXTRA using Catalyst [20] obtains the (batch) optimal $O(\sqrt{\kappa_b}(1 + \tau/\sqrt{\gamma}) \log(\varepsilon^{-1}))$ rate up to log factors. Another line of work on decentralized algorithms is based on the *penalty method* [19, 9]. This consists in performing traditional optimization algorithms to problems augmented with a Laplacian penalty, and in particular enables the use of accelerated methods. Yet, these algorithms are sensitive to the value of the penalty parameter (when it is fixed), since it directly influences the solution they converge to. Another natural way to construct decentralized optimization algorithms is through dual approaches [32, 38]. Although the dual approach leads to algorithms that are optimal both in terms of number of communications and computations [31, 13], they generally assume access to the proximal operator or the gradient of the Fenchel conjugate of the local functions, which is not very practical in general since it requires solving a subproblem at each step.

**Decentralized stochastic optimization.** Although both stochastic and decentralized methods have a rich litterature, there exist few decentralized stochastic methods with linear convergence rate. Although DSA [27], or GT-SAGA [41] propose such algorithms, they respectively take time $O((m\kappa_s + \kappa_s^4 \gamma^{-1}(1 + \tau) \log(\varepsilon^{-1}))$ and $O((m + \kappa_s^2 \gamma^{-2})(1 + \tau) \log(\varepsilon^{-1}))$ to reach precision $\varepsilon$. Therefore, they have significantly worse rates than decentralized batch methods when $m = 1$, and than single-machine stochastic methods when $n = 1$. Other methods have better rates of convergence [36, 12] but they require evaluation of proximal operators, which may be expensive.

**Our contributions.** This work develops a dual approach similar to that of [12], which leads to a decentralized stochastic algorithm with rate $O(m + \kappa_s + \tau\kappa_b/\sqrt{\gamma})$, where the $\sqrt{\gamma}$ factor comes from Chebyshev acceleration, such as used in [32]. Yet, our algorithm, called DVR, can be formulated in the primal only, thus avoiding the need for computing expensive dual gradients or proximal operators. Besides, DVR is derived by applying Bregman coordinate descent to the dual of a specific augmented problem. Thus, its convergence follows from the convergence of block coordinate descent with Bregman gradients, which we prove as a side contribution. When executed on a single-machine, DVR is similar to dual-free SDCA [34], and obtains similar rates. We believe that the same methodology could be applied to tackle non-convex problems, but we leave these extensions for future work.

We present in Section 2 the derivations leading to DVR, namely the dual approach and the dual-free trick. Then, Section 3 presents the actual algorithm along with a convergence theorem based on block Bregman coordinate descent (presented in Appendix A). Section 4 shows how to accelerate DVR, both in terms of network dependence (Chebyshev acceleration) and global iteration complexity (Catalyst acceleration [23]). Finally, experiments on real-world data are presented in Section 5, that demonstrate the effectiveness of DVR.

## 2  Algorithm Design

This section presents the key steps leading to DVR. We start by introducing a relevant dual formulation from [12], then introduce the dual-free trick based on [17], and finally show how this leads to DVR, an actual implementable decentralized stochastic algorithm, as a special case of the previous derivations.

## 2.1  Dual formulation

The standard dual formulation of Problem (1) is obtained by associating a parameter vector to each node, and imposing that two neighboring nodes have the same parameters [6, 15, 32]. This leads to the following constrained problem, in which we write $\theta^{(i)} \in \mathbb{R}^d$ the local vector of node $i$:

$$\min_{\theta \in \mathbb{R}^{nd}} \sum_{i=1}^n f_i(\theta^{(i)}) \text{ such that } \forall k, \ell \in \mathcal{N}(k),\ \theta^{(k)} = \theta^{(\ell)}. \tag{2}$$

Following the approach of [12, 13], we further split the $f_i(\theta^{(i)})$ term into $\sigma_i \|\theta^{(i)}\|^2/2 + \sum_{j=1}^n f_{ij}(\theta^{(ij)})$, with the constraint that $\theta^{(i)} = \theta^{(ij)}$ for all $j$. This is equivalent to the previous approach performed on an augmented graph [12, 13] in which each node is split into a star network with the regularization in the center and a local summand at each tip of the star. Thus, the equivalent augmented constrained problem that we consider writes:

$$\min_{\theta \in \mathbb{R}^{n(m+1)d}} \sum_{i=1}^n \left[ \frac{\sigma_i}{2} \|\theta^{(i)}\|^2 + \sum_{j=1}^m f_{ij}(\theta^{(ij)}) \right] \text{ s.t. } \forall k, \ell \in \mathcal{N}(k),\ \theta^{(k)} = \theta^{(\ell)} \text{ and } \forall i, j,\ \theta^{(i)} = \theta^{(ij)}. \tag{3}$$

We now use Lagrangian duality, and introduce two kinds of multipliers. The variable $x$ corresponds to multipliers associated with the constraints given by edges of the communication graph (*i.e.*, $\theta^{(k)} = \theta^{(\ell)}$ if $k \in \mathcal{N}(\ell)$), that we will call *communication edges*. Similarly, $y$ corresponds to the constraints associated with the edges that are specific to the augmented graph (*i.e.*, $\theta^{(i)} = \theta^{(ij)} \, \forall i, j$) that we call *computation* or *virtual edges*, since they are not present in the original graph and were constructed for the augmented problem. Therefore, there are $E$ communication edges (number of edges in the initial graph), and $nm$ virtual edges. The dual formulation of Problem (3) thus writes:

$$\min_{x \in \mathbb{R}^{Ed},\ y \in \mathbb{R}^{nmd}} \frac{1}{2} q_A(x, y) + \sum_{i=1}^n \sum_{j=1}^m f_{ij}^*((A(x,y))^{(ij)}), \text{ with } q_A(x, y) \triangleq (x, y)^\top A^\top \Sigma A(x, y), \tag{4}$$

and where $(x, y) \in \mathbb{R}^{(E+nm)d}$ is the concatenation of vectors $x \in \mathbb{R}^{Ed}$, which is associated with the communication edges, and $y \in \mathbb{R}^{nmd}$, which is the vector associated with computation edges. We denote $\Sigma = \text{Diag}(\sigma_1^{-1}, \cdots, \sigma_n^{-1}, 0, \cdots, 0) \otimes I_d \in \mathbb{R}^{n(m+1)d \times n(m+1)d}$ and $A$ is such that for all $z \in \mathbb{R}^d$, $A(e_{k,\ell} \otimes z) = \mu_{k\ell}(u_k - u_\ell) \otimes P_{k\ell}z$ for edge $(k, \ell)$, where $P_{k\ell} = I_d$ if $(k, \ell)$ is a communication edge, $P_{ij}$ is the projector on $\text{Ker}(f_{ij})^\perp \triangleq (\cap_{x \in \mathbb{R}^d} \text{Ker}(\nabla^2 f_{ij}(x)))^\perp$ if $(i, j)$ is a virtual edge, $z_1 \otimes z_2$ is the Kronecker product of vectors $z_1$ and $z_2$, and $e_{k,\ell} \in \mathbb{R}^{E+nm}$ and $u_k \in R^{n(m+1)}$ are the unit vectors associated with edge $(k, \ell)$ and node $k$ respectively.

Note that the upper left $nd \times nd$ block of $AA^\top$ (corresponding to the communication edges) is equal to $W \otimes I_d$ where $W$ is a gossip matrix (see, *e.g.*, [32]) that depends on the $\mu_{k\ell}$. In particular, $W$ is equal to the Laplacian of the communication graph if $\mu_{k\ell}^2 = 1/2$ for all $(k, \ell)$. For computation edges, the projectors $P_{ij}$ account for the fact that the parameters $\theta^{(i)}$ and $\theta^{(ij)}$ only need to be equal on the subspaces on which $f_{ij}$ is not constant, and we choose $\mu_{ij}$ such that $\mu_{ij}^2 = \alpha L_{ij}$ for some $\alpha > 0$. Although this introduces heavier notations, explicitly writing $A$ as an $n(1+m)d \times (E+nm)d$ matrix instead of an $n(1+m) \times (E+nm)$ matrix allows to introduce the projectors $P_{ij}$, which then yields a better communication complexity than choosing $P_{ij} = I_d$. See [12, 13] for more details on this dual formulation, and in particular on the construction on the augmented graph. Now that we have obtained a suitable dual problem, we would like to solve it without computing gradients or proximal operators of $f_{ij}^*$, which can be very expensive.

## 2.2  Dual-free trick

Dual methods are based on variants of Problem (4), and apply different algorithms to it. In particular, [32, 38] use accelerated gradient descent [30], and [11, 12] use accelerated (proximal) coordinate descent [24]. Let $p_{\text{comm}}$ denote the probability of performing a communication step and $p_{ij}$ be the probability that node $i$ samples a gradient of $f_{ij}$, which are such that for all $i$, $\sum_{j=1}^m p_{ij} = 1 - p_{\text{comm}}$. Applying a coordinate update with step-size $\eta/p_{\text{comm}}$ to Problem (4) in the direction $x$ (associated with communication edges) writes:

$$x_{t+1} = x_t - \eta p_{\text{comm}}^{-1} \nabla_x q_A(x_t, y_t), \tag{5}$$

where we denote $\nabla_x$ the gradient in coordinates that correspond to $x$ (communication edges), and $\nabla_{y,ij}$ the gradient for coordinate $(ij)$ (computation edge). Similarly, the standard coordinate update of a local computation edge $(i,j)$ can be written as:

$$y_{t+1}^{(ij)} = \arg\min_{y \in \mathbb{R}^d} \left\{ \left( \nabla_{y,ij} q_A(x_t, y_t) + \mu_{ij} \nabla f_{ij}^*(\mu_{ij} y_t^{(ij)}) \right)^\top y + \frac{p_{ij}}{2\eta} \|y - y_t^{(ij)}\|^2 \right\}, \quad (6)$$

where the minimization problem actually has a closed form solution. Yet, as mentioned before, solving Equation (6) requires computing the derivative of $f_{ij}^*$. In order to avoid this, a trick introduced by [17] and later used in [39] is to replace the Euclidean distance term by a well-chosen Bregman divergence. More specifically, the Bregman divergence of a convex function $\phi$ is defined as:

$$D_\phi(x, y) = \phi(x) - \phi(y) - \nabla\phi(y)^\top(x - y). \quad (7)$$

Bregman gradient algorithms typically enjoy the same kind of guarantees as standard gradient algorithms, but with slightly different notions of *relative* smoothness and strong convexity [1, 25]. Note that the Bregman divergence of the squared Euclidean norm is the squared Euclidean distance, and the standard gradient descent algorithm is recovered in that case. We now replace the Euclidean distance by the Bregman divergence induced by function $\phi : y \mapsto (L_{ij}/\mu_{ij}^2) f_{ij}^*(\mu_{ij} y^{(ij)})$, which is normalized to be 1-strongly convex since $f_{ij}^*$ is $L_{ij}^{-1}$-strongly convex. We introduce the constant $\alpha > 0$ such that $\mu_{ij}^2 = \alpha L_{ij}$ for all computation edges $(i,j)$. Using the definition of the Bregman divergence with respect to $\phi$, we write:

$$
\begin{aligned}
y_{t+1}^{(ij)} &= \arg\min_{y \in \mathbb{R}^d} \left( \nabla_{y,ij} q_A(x_t, y_t) + \mu_{ij} \nabla f_{ij}^*(\mu_{ij} y_t^{(ij)}) \right)^\top y + \frac{p_{ij}}{\eta} D_\phi \left( y, y_t^{(ij)} \right) \\
&= \arg\min_{y \in \mathbb{R}} \left( \frac{\alpha\eta}{p_{ij}} \nabla_{y,ij} q_A(x_t, y_t) - \left( 1 - \frac{\alpha\eta}{p_{ij}} \right) \mu_{ij} \nabla f_{ij}^*(\mu_{ij} y_t^{(ij)}) \right)^\top y + f_{ij}^*(\mu_{ij} y) \\
&= \frac{1}{\mu_{ij}} \nabla f_{ij} \left( \left( 1 - \frac{\alpha\eta}{p_{ij}} \right) \nabla f_{ij}^*(\mu_{ij} y_t^{(ij)}) - \frac{\alpha\eta}{\mu_{ij} p_{ij}} \nabla_{y,ij} q_A(x_t, y_t) \right).
\end{aligned}
$$

In particular, if we know $\nabla f_{ij}^*(\mu_{ij} y_t^{(ij)})$ then it is possible to compute $y_{t+1}^{(ij)}$. Besides,

$$\nabla f_{ij}^*(\mu_{ij} y_{t+1}^{(ij)}) = (1 - \alpha\eta) \nabla f_{ij}^*(\mu_{ij} y_t^{(ij)}) - \frac{\alpha\eta}{\mu_{ij}} \nabla_{y,ij} q_A(x_t, y_t), \quad (8)$$

so we can also compute $\nabla f_{ij}^*(\mu_{ij} y_{t+1}^{(ij)})$, and we can use it for the next step. Therefore, instead of computing a dual gradient at each step, we can simply choose $y_0^{(i)} = \mu_{ij}^{-1} \nabla f_{ij}(z_0^{(ij)})$ for any $z_0^{(ij)}$, and iterate from this. Therefore, the Bregman coordinate update applied to Problem (4) in the block of direction $(i,j)$ with $y_0^{(ij)} = \mu_{ij}^{-1} \nabla f_i(z_0^{(ij)})$ yields:

$$z_{t+1}^{(ij)} = \left( 1 - \frac{\alpha\eta}{p_{ij}} \right) z_t^{(ij)} - \frac{\alpha\eta}{p_{ij}\mu_{ij}} \nabla_{y,ij} q_A(x_t, y_t), \qquad y_{t+1}^{(ij)} = \mu_{ij}^{-1} \nabla f_i(z_{t+1}^{(ij)}). \quad (9)$$

The iterations of (9) are called a *dual-free* algorithm because they are a transformation of the iterations from (6) that do not require computing $\nabla f_{ij}^*$ anymore. This is obtained by replacing the Euclidean distance in (6) by the Bregman divergence of a function proportional to $f_{ij}^*$. Note that although we use the same dual-free trick the tools are different since [17] applies a randomized primal-dual algorithm with fixed Bregman divergences choice to a specific *primal-dual* formulation. Instead, we apply a generic randomized Bregman coordinate descent algorithm to a specific *dual* formulation.

## 2.3 Distributed implementation

Iterations from (9) do not involve functions $f_{ij}^*$ anymore, which was our first goal. Yet, they consist in updating dual variables associated with edges of the augmented graph, and have no clear distributed meaning yet. In this section, we rewrite the updates of (9) in order to have an easy to implement distributed algorithm. The key steps are (i) multiplication of the updates by $A$, (ii) expliciting the gossip matrix and (iii) remarking that $\theta_t^{(i)} = (\Sigma A(x_t, y_t))^{(i)}$ converges to the primal solution for all $i$. For a vector $z \in \mathbb{R}^{(n+nm)d}$, we denote $[z]_{\text{comm}} \in \mathbb{R}^{nd}$ its restriction to the communication nodes, and $[M]_{\text{comm}} \in \mathbb{R}^{nd \times nd}$ similarly refers to the restriction on communication edges of a matrix

$M \in \mathbb{R}^{(n+nm)d \times (n+nm)d}$. By abuse of notations, we call $A_{\text{comm}} \in \mathbb{R}^{nd \times Ed}$ the restriction of $A$ to communication nodes and edges. We denote $P_{\text{comm}}$ the projector on communication edges, and $P_{\text{comp}}$ the projector on $y$. We multiply the $x$ (communication) update in (9) by $A$ on the left (which is standard [32, 12]) and obtain:

$$A_{\text{comm}}x_{t+1} = A_{\text{comm}}x_t - \eta p_{\text{comm}}^{-1}[AP_{\text{comm}}A^\top]_{\text{comm}}[\Sigma A(x_t, y_t)]_{\text{comm}}. \tag{10}$$

Note that $[P_{\text{comm}}A^\top \Sigma A(x_t, y_t)]_{\text{comm}} = [P_{\text{comm}}A^\top]_{\text{comm}}[\Sigma A(x_t, y_t)]_{\text{comm}}$ because $P_{\text{comm}}$ and $\Sigma$ are non-zero only for communication edges and nodes. Similarly, and as previously stated, one can verify that $A_{\text{comm}}[P_{\text{comm}}A^\top]_{\text{comm}} = [AP_{\text{comm}}A^\top]_{\text{comm}} = W \otimes I_d \in \mathbb{R}^{nd \times nd}$ where $W$ is a gossip matrix. We finally introduce $\tilde{x}_t \in \mathbb{R}^{nd}$ which is a variable associated with nodes, and which is such that $\tilde{x}_t = A_{\text{comm}}x_t$. With this rewriting, the communication update becomes:

$$\tilde{x}_{t+1} = \tilde{x}_t - \eta p_{\text{comm}}^{-1}(W \otimes I_d)\Sigma_{\text{comm}}\left[A(x_t, y_t)\right]_{\text{comm}}.$$

To show that $[A(x_t, y_t)]_{\text{comm}}$ is locally accessible to each node, we write:

$$[A(x_t, y_t)]_{\text{comm}}^{(i)} = (A_{\text{comm}}x_t)^{(i)} - \left(\sum_{k=1}^{n}\sum_{j=1}^{m}(A(e_{kj} \otimes y_t^{(kj)}))^{(i)}\right) = (\tilde{x}_t)^{(i)} - \sum_{j=1}^{m}\mu_{ij}y_t^{(ij)}.$$

We note this rescaled local vector $\theta_t = \Sigma_{\text{comm}}([A(x_t, y_t)]_{\text{comm}})$, and obtain for variables $\tilde{x}_t$ the gossip update of (12). Note that we directly write $y_t^{(ij)}$ instead of $P_{ij}y_t^{(ij)}$ even though there has been a multiplication by the matrix $A$. This is allowed because Equation (13) implies that (i) $y_t^{(ij)} \in \text{Ker}(f_{ij})^\perp$ for all $t$, and (ii) the value of $(I_d - P_{ij})z_t^{(ij)}$ does not matter since $z_t^{(ij)}$ is only used to compute $\nabla f_{ij}$. We now consider computation edges, and remark that:

$$\nabla_{y,ij}q_A(x_t, y_t) = -\mu_{ij}(\Sigma_{\text{comm}})_{ii}([A(x_t, y_t)]_{\text{comm}})^{(i)} = -\mu_{ij}\theta_t. \tag{11}$$

Plugging Equation (11) into the updates of (9), we obtain the following updates:

$$\tilde{x}_{t+1} = \tilde{x}_t - \frac{\eta}{p_{\text{comm}}}(W \otimes I_d)\theta_t, \tag{12}$$

for communication edges, and for the local update of the $j$-th component of node $i$:

$$z_{t+1}^{(ij)} = \left(1 - \frac{\alpha\eta}{p_{ij}}\right)z_t^{(ij)} + \frac{\alpha\eta}{p_{ij}}\theta_t^{(i)}, \qquad \theta_{t+1}^{(i)} = \frac{1}{\sigma_i}\left(\tilde{x}_{t+1}^{(i)} - \sum_{j=1}^{m}\nabla f_{ij}(z_{t+1}^{(ij)})\right). \tag{13}$$

Finally, Algorithm 1 is obtained by expressing everything in terms of $\theta_t$ and removing variable $\tilde{x}_t$. To simplify notations, we further consider $\theta$ as a matrix in $\mathbb{R}^{n \times d}$ (instead of a vector in $\mathbb{R}^{nd}$), and so the communication update of Equation (12) is a standard gossip update with matrix $W$, which we recall is such that $W \otimes I_d = [AP_{\text{comm}}A^\top]_{\text{comm}}$. We now discuss the local updates of Equation (13) more in details, which are closely related to dual-free SDCA updates [34].

## 3 Convergence Rate

The goal of this section is to set parameters $\eta$ and $\alpha$ in order to get the best convergence guarantees. We introduce $\kappa_{\text{comm}} = \gamma\lambda_{\max}(A_{\text{comm}}^\top \Sigma_{\text{comm}}A_{\text{comm}})/\lambda_{\min}^+(A_{\text{comm}}^\top D_M^{-1}A_{\text{comm}})$, where $\lambda_{\min}^+$ and $\lambda_{\max}$ respectively refer to the smallest non-zero and the highest eigenvalue of the corresponding matrices. We denote $D_M$ the diagonal matrix such that $(D_M)_{ii} = \sigma_i + \lambda_{\max}(\sum_{j=1}^{m}L_{ij}P_{ij})$, where $\nabla^2 f_{ij}(x) \preccurlyeq L_{ij}P_{ij}$ for all $x \in \mathbb{R}^d$. Note that we use notation $\kappa_{\text{comm}}$ since it corresponds to a condition number. In particular, $\kappa_{\text{comm}} \leq \kappa_s$ when $\sigma_i = \sigma_j$ for all $i, j$, and $\kappa_{\text{comm}}$ more finely captures the interplay between regularity of local functions (through $D_M$ and $\Sigma_{\text{comm}}$) and the topology of the network (through $A$) otherwise.

**Theorem 1.** *We choose $p_{\text{comm}} = \left(1 + \gamma\frac{m+\kappa_s}{\kappa_{\text{comm}}}\right)^{-1}$, $p_{ij} \propto (1 - p_{\text{comm}})(1 + L_{ij}/\sigma_i)$ and $\alpha$ and $\eta$ as in Algorithm 1. Then, there exists $C_0 > 0$ that only depends on $\theta_0$ (initial conditions) such that for all $t > 0$, the error and the expected time $T_\varepsilon$ required to reach precision $\varepsilon$ are such that:*

$$\sum_{i=1}^{n}\frac{1}{2}\mathbb{E}\left[\|\theta_t^{(i)} - \theta^\star\|^2\right] \leq C_0\left(1 - \frac{\alpha\eta}{2}\right)^t, \text{ and so } T_\varepsilon = O\left(\left[m + \kappa_s + \tau\frac{\kappa_{\text{comm}}}{\gamma}\right]\log\varepsilon^{-1}\right).$$

**Algorithm 1** DVR($z_0$)

---

1: $\alpha = 2\lambda_{\min}^+(A_{\text{comm}}^\top D_M^{-1} A_{\text{comm}})$, $\eta = \min\left(\frac{p_{\text{comm}}}{\lambda_{\max}(A_{\text{comm}}^\top \Sigma_{\text{comm}} A_{\text{comm}})}, \frac{p_{ij}}{\alpha(1+\sigma_i^{-1} L_{ij})}\right)$    *// Init.*

2: $\theta_0^{(i)} = -(\sum_{j=1}^m \nabla f_{ij}(z_0^{(ij)}))/\sigma_i.$    *// $z_0$ is arbitrary but not $\theta_0$.*

3: **for** $t = 0$ to $K-1$ **do**    *// Run for $K$ iterations*

4:   Sample $u_t$ uniformly in $[0,1]$.    *// Randomly decide the kind of update*

5:   **if** $u_t \le p_{\text{comm}}$ **then**

6:     $\theta_{t+1} = \theta_t - \frac{\eta}{p_{\text{comm}}} \Sigma W \theta_t$    *// Communication using $W$*

7:   **else**

8:     **for** $i = 1$ to $n$ **do**

9:       Sample $j \in \{1, \cdots, m\}$ with probability $p_{ij}$.

10:       $z_{t+1}^{(ij')} = z_t^{(ij')}$ for $j \ne j'$    *// Only one virtual node is updated*

11:       $z_{t+1}^{(ij)} = \left(1 - \frac{\alpha\eta}{p_{ij}}\right) z_t^{(ij)} + \frac{\alpha\eta}{p_{ij}} \theta_t^{(i)}$    *// Virtual node update*

12:       $\theta_{t+1}^{(i)} = \theta_t^{(i)} - \frac{1}{\sigma_i}\left(\nabla f_{ij}(z_{t+1}^{(ij)}) - \nabla f_{ij}(z_t^{(ij)})\right)$    *// Local update using $f_{ij}$*

13: **return** $\theta_K$

---

*Proof sketch.* We have seen in Section 2 that DVR is obtained by applying Bregman coordinate descent on a well-chosen dual problem. Therefore, one of our key results consists in proving convergence rates for Bregman coordinate descent in the relatively smooth setting. Although a similar algorithm is analyzed in [10], we give sharper results in the case of arbitrary sampling of blocks, and tightly adapt to the separability structure. This is crucial to our analysis since the probabilities to sample a local gradient and to communicate can be vastly different. In order to ease the reading of the paper, we present these results for a general setting in Appendix A, which is self-contained and which we believe to be of independent interest (beyond its application to decentralized optimization).

Then, Appendix B focuses on the application to decentralized optimization. In particular, we recall the Equivalence between DVR and Bregman coordinate descent applied to the dual problem of Equation (4), and show that its structure is suited to the application of coordinate descent. Indeed, no two virtual edges adjacent to the same node are updated at the same time with our sampling. Then, we evaluate the relative smoothness and strong convexity constants of the augmented problem, which is rather challenging due to the complex structure of the dual problem. This allows to derive adequate values for parameters $\alpha$ and $\eta$. Finally, we choose $p_{\text{comm}}$ in order to minimize the execution time of DVR. □

We would like to highlight the fact that the convergence theory of DVR decomposes nicely into several building blocks, and thus simple rates are obtained. This is not so usual for decentralized algorithms, for instance many follow-up papers were needed to obtain a tight convergence theory for EXTRA [37, 14, 42, 20]. We now discuss the convergence rate of DVR more in details.

**Computation complexity.** The computation complexity of DVR is the same computation complexity as locally running a stochastic algorithm with variance reduction at each node. This is not surprising since, as we argue later, DVR can be understood as a decentralized version of an algorithm that is closely related to dual-free SDCA [34]. Therefore, this improves the computation complexity of EXTRA from $O(m(\kappa_b+\gamma^{-1}))$ individual gradients to $O(m+\kappa_s)$, which is the expected improvement for stochastic variance-reduced algorithm. In comparison, GT-SAGA [41], a recent decentralized stochastic algorithm, has a computation complexity of order $O(m + \kappa_s^2/\gamma^2)$, which is significantly worse than that of DVR, and generally worse than that of EXTRA as well.

**Communication complexity.** The communication complexity of DVR (*i.e.*, the number of communications, so the communication time is retrieved by multiplying by $\tau$) is of order $O(\kappa_{\text{comm}}/\gamma)$, and can be improved to $O(\kappa_{\text{comm}}/\sqrt{\gamma})$ using Chebyshev acceleration (see Section 4). Yet, this is in general worse than the $O(\kappa_b + \gamma^{-1})$ communication complexity of EXTRA or NIDS, which can be interpreted as a partly accelerated communication complexity since the optimal dependence is $O(\sqrt{\kappa_b/\gamma})$ [31], and $2\sqrt{\kappa_b/\gamma} = \kappa_b + \gamma^{-1}$ in the worst case ($\kappa_b = \gamma^{-1}$). Yet, stochastic updates are mainly intended to deal with cases in which the computation time dominates, and we show in the experimental section that DVR outperforms EXTRA and NIDS for a wide range of communication

times $\tau$ (the computation complexity dominates roughly as long as $\tau < \sqrt{\gamma}(m + \kappa_s)/\kappa_{\text{comm}}$). Finally, the communication complexity of DVR is significantly lower than that of DSA and GT-SAGA, the primal decentralized stochastic alternatives presented in Section 1.

**Homogeneous parameter choice.** In the homogeneous case ($\sigma_i = \sigma_j$ for all $i, j$), choosing the optimal $p_{\text{comp}}$ and $p_{\text{comm}}$ described above leads to $\eta \lambda_{\max}(W) = \sigma p_{\text{comm}}$. Therefore, the communication update becomes $\theta_{t+1} = (I - W/\lambda_{\max}(W)) \theta_t$, which is a gossip update with a standard step-size (independent of the optimization parameters). Similarly, $\alpha\eta(m + \kappa_s) = p_{\text{comp}}$, and so the step-size for the computation updates is independent of the network.

**Links with SDCA.** The single-machine version of Algorithm 1 ($n = 1$, $p_{\text{comm}} = 0$) is closely related to dual-free SDCA [34]. The difference is in the stochastic gradient used: DVR uses $\nabla f_{ij}(z_t^{(ij)})$, where $z_t^{(ij)}$ is a convex combination of $\theta_k^{(i)}$ for $k < t$, whereas dual-free SDCA uses $g_t^{(ij)}$, which is a convex combination of $\nabla f_{ij}(\theta_k^{(i)})$ for $k < t$. Both algorithms obtain the same rates.

**Local synchrony.** Instead of using the synchronous communications of Algorithm 1, it is possible to update edges one at a time, as in [12]. This can be very efficient in heterogeneous settings (both in terms of computation and communication times) and similar convergence results can be obtained using the same framework, and we leave the details for future work.

## 4 Acceleration

We show in this section how to modify DVR to improve the convergence rate of Theorem 1.

**Network acceleration.** Algorithm 1 depends on $\gamma^{-1}$, also called the *mixing time* of the graph, which can be as high as $O(n^2)$ for a chain of length $n$ [26]. However, it is possible to improve this dependency to $\gamma^{-1/2}$ by using Chebyshev acceleration, as in [32]. To do so, the first step is to choose a polynomial $P$ of degree $k$ and communicate with $P(W)$ instead of $W$. In terms of implementation, this comes down to performing $k$ communication rounds instead of one, but this makes the algorithm depend on the spectral gap of $P(W)$. Then, the important fact is that there is a polynomial $P_\gamma$ of degree $\lceil \gamma^{-1/2} \rceil$ such that the spectral gap of $P_\gamma(W)$ is of order 1. Each communication step with $P_\gamma(W)$ only takes time $\tau \deg(P_\gamma) = \tau \lceil \gamma^{-1/2} \rceil$, and so the communication term in Theorem 1 can be replaced by $\tau \kappa_{\text{comm}} \gamma^{-1/2}$, thus leading to *network acceleration*. The polynomial $P_\gamma$ can for example be chosen as a Chebyshev polynomial, and we refer the interested reader to [32] for more details. Finally, other polynomials yield even faster convergence when the graph topology is known [2].

**Catalyst acceleration.** Catalyst [22] is a generic framework that achieves acceleration by solving a sequence of subproblems. Because of space limitations, we only present the accelerated convergence rate without specifying the algorithm in the main text. Yet, only mild modifications to Algorithm 1 are required to obtain these rates, and the detailed derivations and proofs are presented in Appendix C.

**Theorem 2.** *DVR can be accelerated using catalyst, so that the time $T_\varepsilon$ required to reach precision $\varepsilon$ is equal (up to log factors) to*

$$T_\varepsilon = \tilde{O}\left(\left[m + \sqrt{m\kappa_s} + \tau\sqrt{\frac{\kappa_{\text{comm}}}{\gamma}} \times \sqrt{m\frac{\kappa_{\text{comm}}}{\kappa_s}}\right] \log \varepsilon^{-1}\right)$$

*Proof sketch.* We follow the approach of [20] to derive the algorithm, and apply Catalyst acceleration to the primal problem on the mean parameter $\bar{\theta}_t$ (which is never explicitly computed). Indeed, this conceptual algorithm can actually be implemented in a fully decentralized manner.

Then, we proceed to the actual proof, which requires a tight control over both primal and dual warm-start errors. Indeed, Theorem 4 (Appendix B) controls dual variables but Catalyst acceleration is applied to the primal variables. □

This rate recovers the computation complexity of optimal finite sum algorithms such as ADFS [12, 13]. Although the communication time is slightly increased (by a factor $\sqrt{m\kappa_{\text{comm}}/\kappa_s}$), ADFS uses a stronger oracle than DVR (proximal operator instead of gradient), which is why we develop DVR in the first place. Although both ADFS and DVR are derived using the same dual formulation, both the approach and the resulting algorithms are rather different: ADFS uses accelerated coordinate

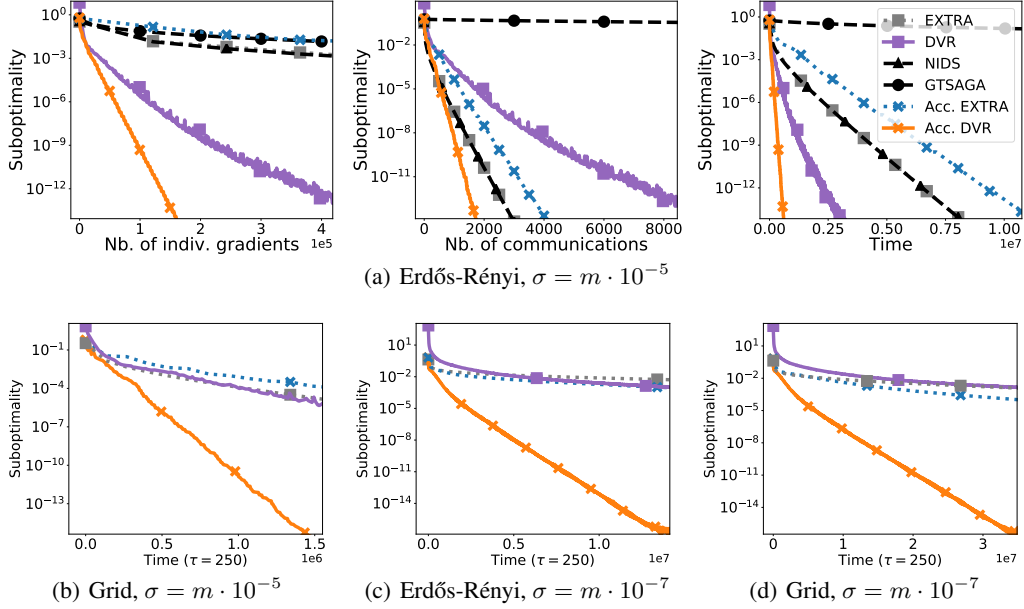

(a) Erdős-Rényi, $\sigma = m \cdot 10^{-5}$

(b) Grid, $\sigma = m \cdot 10^{-5}$    (c) Erdős-Rényi, $\sigma = m \cdot 10^{-7}$    (d) Grid, $\sigma = m \cdot 10^{-7}$

Figure 1: Experimental results for the RCV1 dataset with different graphs of size $n = 81$, with $m = 2430$ samples per node, and with different regularization parameters.

descent, and thus has strong convergence guarantees at the cost of requiring dual oracles. DVR uses coordinate descent with the Bregman divergence of $\phi_{ij} \propto f_{ij}^*$ in order to work with primal oracles, but thus loses direct acceleration, which is recovered through the Catalyst framework. Note that the parameters of accelerated DVR can also be set such that $T_\varepsilon = \tilde{O}\left(\sqrt{\kappa_{\text{comm}}} \left[m + \tau/\sqrt{\gamma}\right] \log \varepsilon^{-1}\right)$, which recovers the convergence rate of optimal batch algorithms, but loses the finite-sum speedup.

## 5 Experiments

We investigate in this section the practical performances of DVR. We solve a regularized logistic regression problem on the RCV1 dataset [18] ($d = 47236$) with $n = 81$ (leading to $m = 2430$) and two different graph topologies: an Erdős-Rényi random graph (see, *e.g.*, [3]) and a grid. We choose $\mu_{k\ell}^2 = 1/2$ for all communication edges, so the gossip matrix $W$ is the Laplacian of the graph.

Figure 1 compares the performance of DVR with that of state-of-the-art primal algorithms such as EXTRA [37], NIDS [21], GT-SAGA [41], and Catalyst accelerated versions of EXTRA [20] and DVR. Suboptimality refers to $F(\theta_t^{(0)}) - F(\theta^\star)$, where node 0 is chosen arbitrarily and $F(\theta^\star)$ is approximated by the minimal error over all iterations. Each subplot of Figure 1(a) shows the same run with different x axes. The left plot measures the complexity in terms of individual gradients ($\nabla f_{ij}$) computed by each node whereas the center plot measures it in terms of communications (multiplications by $W$). All other plots are taken with respect to (simulated) time (*i.e.*, computing $\nabla f_{ij}$ takes time 1 and multiplying by $W$ takes time $\tau$) with $\tau = 250$ in order to report results that are independent of the computing cluster hardware and status. All parameters are chosen according to theory, except for the smoothness of the $f_i$, which requires finding the smallest eigenvalue of a $d \times d$ matrix. For this, we start with $L_b = \sigma_i + \sum_{j=1}^{m} L_{ij}$ (which is a known upper bound), and decrease it while convergence is ensured, leading to $\kappa_b = 0.01\kappa_s$. The parameters for accelerated EXTRA are chosen as in [20] since tuning the number of inner iterations does not significantly improve the results (at the cost of a high tuning effort). For accelerated DVR, we set the number of inner iterations to $N/p_{\text{comp}}$ (one pass over the local dataset). We use Chebyshev acceleration for (accelerated) DVR but not for (accelerated) EXTRA since it is actually slower, as predicted by the theory.

As expected from their theoretical iteration complexities, NIDS and EXTRA perform very similarly [20], and GT-SAGA is the slowest method. Therefore, we only plot NIDS and GT-SAGA in Figure 1(a). We then see that though it requires more communications, DVR has a much lower

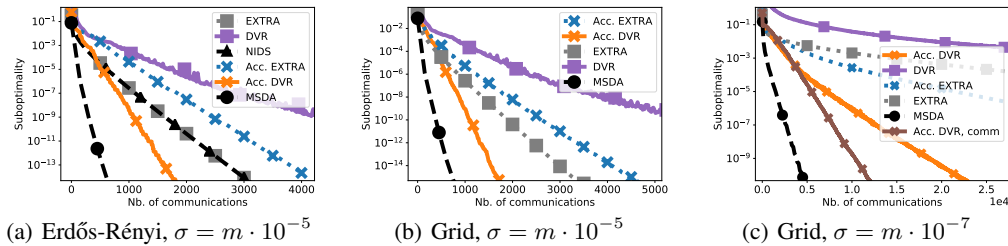

(a) Erdős-Rényi, $\sigma = m \cdot 10^{-5}$    (b) Grid, $\sigma = m \cdot 10^{-5}$    (c) Grid, $\sigma = m \cdot 10^{-7}$

Figure 2: Experimental results for the RCV1 dataset with different graphs of size $n = 81$, with $m = 2430$ samples per node, and with different regularization parameters.

computation complexity than EXTRA, which illustrates the benefits of stochastic methods. We see that DVR is faster overall if we choose $\tau = 250$, and both methods perform similarly for $\tau \approx 1000$, at which point communicating takes roughly as much time as computing a full local gradient. We then see that accelerated EXTRA has quite a lot of overhead and, despite our tuning efforts, is slower than EXTRA when the regularization is rather high. On the other hand, accelerated DVR consistently outperforms DVR by a relatively large margin. The communication complexity is in particular greatly improved, allowing accelerated DVR to be the fastest method regardless of the setting.

Finally, Figure 2 presents the comparison between DVR and MSDA [32], an optimal decentralized batch algorithm, in terms of communication complexity. To implement MSDA, we compute the dual gradients by solving each local subproblem ($\nabla f^*(x) = \arg\max_y x^\top y - f(y)$) up to precision $10^{-11}$ using accelerated gradient descent. Solving the subproblems with lower precision caused MSDA to plateau and not converge to the true optimum. In Figure 2(c), *Acc. DVR comm* (the brown line) refers to Accelerated DVR with Catalyst parameter chosen to favor communication complexity (as explained after Theorem 2). MSDA is the fastest algorithm as expected, but accelerated DVR is not too far behind, especially given the fact that it relies on generic Catalyst acceleration, which adds some complexity overhead. Therefore, the comparison with MSDA corroborates the fact that accelerated DVR is competitive with optimal methods in terms of communication while enjoying a drastically lower computational cost. Further experimental results are given in Appendix D, and the code is available in supplementary material and at `https://github.com/HadrienHx/DVR_NeurIPS`.

## 6   Conclusion

This paper introduces DVR, a Decentralized stochastic algorithm with Variance Reduction obtained using Bregman block coordinate descent on a well-chosen dual formulation. Thanks to this approach, DVR inherits from the fast rates and simple theory of dual approaches without the computational burden of relying on dual oracles. Therefore, DVR has a drastically lower computational cost than standard primal decentralized algorithms, although sometimes at the cost of a slight increase in communication complexity. The framework used to derive DVR is rather general and could in particular be extended to analyze asynchronous algorithms. Finally, although deriving a direct acceleration of DVR is a challenging open problem, Catalyst and Chebyshev accelerations allow to significantly reduce DVR's communication overhead both in theory and in practice.

### Acknowledgement

This work was funded in part by the French government under management of Agence Na-tionalede la Recherche as part of the "Investissements d'avenir" program, reference ANR-19-P3IA-0001(PRAIRIE 3IA Institute). We also acknowledge support from the European Research Council (grantSEQUOIA 724063) and from the MSR-INRIA joint centre.

### Broader impact statement

This work does not present any foreseeable societal consequence.

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
