[Supplementary Material]

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

_{\mathrm{comm}}x_{t+1} = A_{\mathrm{comm}}x_t - \eta p_{\mathrm{comm}}^{-1}[AP_{\mathrm{comm}}A^\top]_{\mathrm{comm}}[\Sigma A(x_t, y_t)]_{\mathrm{comm}}. \tag{10}$$

Note that $[P_{\mathrm{comm}}A^\top \Sigma A(x_t, y_t)]_{\mathrm{comm}} = [P_{\mathrm{comm}}A^\top]_{\mathrm{comm}}[\Sigma A(x_t, y_t)]_{\mathrm{comm}}$ because $P_{\mathrm{comm}}$ and $\Sigma$ are non-zero only for communication edges and nodes. Similarly, and as previously stated, one can verify that $A_{\mathrm{comm}}[P_{\mathrm{comm}}A^\top]_{\mathrm{comm}} = [AP_{\mathrm{comm}}A^\top]_{\mathrm{comm}} = W \otimes I_d \in \mathbb{R}^{nd \times nd}$ where $W$ is a gossip matrix. We finally introduce $\tilde{x}_t \in \mathbb{R}^{nd}$ which is a variable associated with nodes, and which is such that $\tilde{x}_t = A_{\mathrm{comm}}x_t$. With this rewriting, the communication update becomes:

$$\tilde{x}_{t+1} = \tilde{x}_t - \eta p_{\mathrm{comm}}^{-1}(W \otimes I_d)\Sigma_{\mathrm{comm}}\left[A(x_t, y_t)\right]_{\mathrm{comm}}.$$

To show that $[A(x_t, y_t)]_{\mathrm{comm}}$ is locally accessible to each node, we write:

$$[A(x_t, y_t)]_{\mathrm{comm}}^{(i)} = (A_{\mathrm{comm}}x_t)^{(i)} - \left(\sum_{k=1}^{n}\sum_{j=1}^{m}(A(e_{kj} \otimes y_t^{(kj)}))^{(i)}\right) = (\tilde{x}_t)^{(i)} - \sum_{j=1}^{m}\mu_{ij}y_t^{(ij)}.$$

We note this rescaled local vector $\theta_t = \Sigma_{\mathrm{comm}}([A(x_t, y_t)]_{\mathrm{comm}})$, and obtain for variables $\tilde{x}_t$ the gossip update of (12). Note that we directly write $y_t^{(ij)}$ instead of $P_{ij}y_t^{(ij)}$ even though there has been a multiplication by the matrix $A$. This is allowed because Equation (13) implies that (i) $y_t^{(ij)} \in \mathrm{Ker}(f_{ij})^\perp$ for all $t$, and (ii) the value of $(I_d - P_{ij})z_t^{(ij)}$ does not matter since $z_t^{(ij)}$ is only used to compute $\nabla f_{ij}$. We now consider computation edges, and remark that:

$$\nabla_{y,ij}q_A(x_t, y_t) = -\mu_{ij}(\Sigma_{\mathrm{comm}})_{ii}([A(x_t, y_t)]_{\mathrm{comm}})^{(i)} = -\mu_{ij}\theta_t. \tag{11}$$

Plugging Equation (11) into the updates of (9), we obtain the following updates:

$$\tilde{x}_{t+1} = \tilde{x}_t - \frac{\eta}{p_{\mathrm{comm}}}(W \otimes I_d)\theta_t, \tag{12}$$

for communication edges, and for the local update of the $j$-th component of node $i$:

$$z_{t+1}^{(ij)} = \left(1 - \frac{\alpha\eta}{p_{ij}}\right)z_t^{(ij)} + \frac{\alpha\eta}{p_{ij}}\theta_t^{(i)}, \qquad \theta_{t+1}^{(i)} = \frac{1}{\sigma_i}\left(\tilde{x}_{t+1}^{(i)} - \sum_{j=1}^{m}\nabla f_{ij}(z_{t+1}^{(ij)})\right). \tag{13}$$

Finally, Algorithm 1 is obtained by expressing everything in terms of $\theta_t$ and removing variable $\tilde{x}_t$. To simplify notations, we further consider $\theta$ as a matrix in $\mathbb{R}^{n \times d}$ (instead of a vector in $\mathbb{R}^{nd}$), and so the communication update of Equation (12) is a standard gossip update with matrix $W$, which we recall is such that $W \otimes I_d = [AP_{\mathrm{comm}}A^\top]_{\mathrm{comm}}$. We now discuss the local updates of Equation (13) more in details, which are closely related to dual-free SDCA updates [34].

## 3    Convergence Rate

The goal of this section is to set parameters $\eta$ and $\alpha$ in order to get the best convergence guarantees. We introduce $\kappa_{\mathrm{comm}} = \gamma\lambda_{\max}(A_{\mathrm{comm}}^\top\Sigma_{\mathrm{comm}}A_{\mathrm{comm}})/\lambda_{\min}^+(A_{\mathrm{comm}}^\top D_M^{-1}A_{\mathrm{

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

This appendix contains the details of the derivations and proofs from the main text. More specifically, Appendix A is a self-contained appendix that specifies the Bregman coordinate descent algorithm and proves its convergence rate. Appendix B focuses on the application of Bregman coordinate descent to the dual problem (relative smoothness and strong convexity constants, sparsity structure), and how to retrieve guarantees on the primal parameters. Appendix C is devoted to presenting the Catalyst acceleration of DVR and proving its convergence speed, and Appendix D details the experimental setting, along with more experiments.

## A    Block Coordinate descent

We focus in this section on the general problem minimizing $f + g$ using coordinate Bregman gradient, where $g$ is separable, *i.e.*, $g(x) = \sum_{i=1}^{d} g_i(x^{(i)})$. This is a self-contained section, and notations may differ from the rest of the paper. In particular, function $f$ is for now arbitrary and not related to $F$ or $f_i$ from Problem (1), and the dimension $d$ is arbitrary as well.

We first precise the blocks sampling rule. More specifically, we define a block $b \subset \{1, \dots, d\}$ as a collection of coordinates, and $\mathcal{B}$ is the set of all blocks that can be chosen for the updates. Then, the algorithm updates each block $b \in \mathcal{B}$ with probability $p(b)$, so that the probability of updating a given coordinate is given by $p_i = \sum_{i \in b} p(b)$. Similarly to individual coordinates, we write $x^{(b)}$ the restriction of $x$ to coordinates in $b$. The Bregman coordinate gradient update for a block of coordinates $b$ writes:

$$x_{t+1} = \arg \min_{x \in \mathbb{R}^d} \left\{ V_t^b(x) \triangleq \sum_{i \in b} \frac{\eta_t}{p_i} \left[ \nabla_i f(x_t)^\top x + g_i(x^{(i)}) \right] + D_\phi(x, x_t) \right\}, \qquad (14)$$

where $\nabla_i f$ denotes the gradient of $f$ in direction $i$. Note that this update is more general than the one used to derive DVR, for which $g = 0$. In order to derive strong guarantees for this block coordinate descent algorithm, we need to ensure that there is some separability in functions $f$ and $\phi$, and that the block structure is suited to this separability. All the assumptions about the separability structure of $f$, $g$ and $\phi$ are contained in the following assumption.

**Assumption 1** (Separability). *The function $g$ is separable and the function $\phi$ is block-separable for $b$, meaning that for all $b \in \mathcal{B}$, there exist two convex functions $\phi_b$ and $\phi_b^\perp$ such that for all $x$,*

$$\phi(x) = \phi_b(x^{(b)}) + \phi_b^\perp(x - x^{(b)}). \qquad (15)$$

*Besides, for all $b \in \mathcal{B}$, either of the following two hold:*

1. *$\phi$ and $f$ are separable for b*, i.e., *$\phi_b(x^{(b)}) = \sum_{i \in b} \phi_i(x^{(i)})$, and*

$$\sum_{i \in b} [f(x_t + \delta_i e_i) - f(x_t)] = f\left(x_t + \sum_{i \in b} \delta_i e_i\right) - f(x_t).$$

2. *$p_i = p_j$ for all $i, j \in b$.*

If $\phi$ is not block-separable, the support of the Bregman update in direction $b$ may not restricted to $b$. This causes some of the derivations below to fail, which is why we prevent it by assuming that Equation (15) holds.

Then, the first option ensures that within a block, the updates do not affect each other. The function $f$ is not separable, but some directions can be updated independently from others. To have these independent updates, we also need to assume further separability of $\phi$ within the blocks. The second option states that if only block-separability of $\phi$ is assumed then within each block for which $\phi$ and $f$ are not separable, coordinates must be picked with the same probability.

Assumption 1 is a bit technical but we actually require all statements in order to derive DVR. In particular, the first option is verified when updating within the same block virtual edges that are adjacent to different nodes in the dual problem. The second option is verified when picking all communication edges at once within the same block.

Now that we have made assumptions on the structure of $f$, $g$ and $\phi$, we will make assumptions on their regularity. We start by a directional relative smoothness assumption between $f$ and $\phi$, *i.e.*, we

assume that for all $i$, there exists $L_{\text{rel}}^i$ such that for all $\delta > 0$ and $e_i$ the unit vector of direction $i$,

$$D_f(x + \delta e_i, x) \leq L_{\text{rel}}^i D_\phi(x + \delta e_i, x). \tag{16}$$

Similarly, for $\sigma_{\text{rel}} > 0$, $f$ is said to be $\sigma_{\text{rel}}$-strongly convex relatively to $\phi$ if for all $x, y$:

$$D_f(x, y) \geq \sigma_{\text{rel}} D_\phi(x, y). \tag{17}$$

We finally assume that $f$ and $\phi$ are convex (but not necessarily smooth). We can now state the central theorem of this section:

**Theorem 3.** *Let $f$ and $\phi$ be such that $f$ is $L_{\text{rel}}^i$-smooth in direction $i$ and $\sigma_{\text{rel}}$-strongly convex relatively to $\phi$. Denote $p_{\min} = \min_i p_i$, and*

$$L_t = D_\phi(x, x_t) + \frac{\eta_t}{p_{\min}} \left( F(x_t) - F(x) \right).$$

*Then, if the blocks $\mathcal{B}$ respect Assumption 1 (separability) and $\eta_t L_{\text{rel}}^i < p_i$ for all $i$, the Bregman coordinate descent algorithm guarantees for all $x$:*

$$\mathbb{E}\left[L_{t+1}\right] \leq (1 - \eta_t \sigma_{\text{rel}}) L_t.$$

*The same result holds with $L_t' = D_\phi(x, x_t) + \frac{1}{L_{\text{rel}}^{\max}} \left( F(x_t) - F(x) \right)$, where $L_{\text{rel}}^{\max} = \max_i L_{\text{rel}}^i$.*

To prove this theorem, we start by proving the monotonicity of such iterations.

**Lemma 1** (Monotonicity). *We note $\delta_i = e_i^\top (x_{t+1} - x_t) e_i$. If $x_{t+1} = \arg\min_x V_t^b(x)$ then:*

1. *If $\phi$ and $f$ are separable for $b$ then for all $i \in b$, if $\eta_t L_{\text{rel}}^i \leq p_i$ then $F(x_t) \geq F(x_t + \delta_i)$.*

2. *If $p_i = p_j$ for all $i, j \in b$ and $\eta_t L_{\text{rel}}^b \leq p_b$ then $F(x_t) \geq F(x_{t+1})$.*

*Proof.* We start by the first point. If $\phi$ is separable for $b$ then this means that each coordinate is updated independently. By definition of $x_{t+1}^{(i)}$, we have $V_t^b(x_{t+1}^{(b)}) \leq V_t^b(x_t)$. This writes, splitting over each $i$ and using the fact that $D_{\phi_i}(x_t, x_t) = 0$:

$$
\begin{aligned}
g_i(x_t^{(i)}) - g_i(x_{t+1}^{(i)}) &\geq \nabla_i f(x_t)^\top (x_{t+1}^{(i)} - x_t^{(i)}) + \frac{p_i}{\eta} D_{\phi_i}(x_{t+1}^{(i)}, x_t^{(i)}) \\
&= \nabla_i f(x_t)^\top (x_t + \delta_i - x_t) + \frac{p_i}{\eta} D_{\phi_i}(x_{t+1}^{(i)}, x_t^{(i)}) \\
&= f(x_t + \delta_i) - f(x_t) - D_f(x_{t+1}^{(i)}, x_t^{(i)}) + \frac{p_i}{\eta} D_{\phi_i}(x_{t+1}^{(i)}, x_t^{(i)}) \\
&\geq f(x_t + \delta_i) - f(x_t) + \left( \frac{p_i}{\eta} - L_{\text{rel}}^i \right) D_{\phi_i}(x_{t+1}^{(i)}, x_t^{(i)}) \\
&\geq f(x_t + \delta_i) - f(x_t).
\end{aligned}
$$

The result follows from summing over all $i \in b$, and using Assumption 1. For the second point, it is not possible to split the update per coordinate since $\phi$ is not separable. Yet, we can still write (using separability of $g$):

$$\sum_{i \in b} \frac{\eta_t}{p_i} \left[ g(x_t^{(i)}) - g(x_{t+1}^{(i)}) - \nabla_i f(x_t)^\top (x_{t+1}^{(i)} - x_t^{(i)}) \right] \geq D_\phi(x_{t+1}, x_t). \tag{18}$$

Since $g$ is separable and $p_i = p_b$ for all $i \in b$, Equation (18) writes:

$$g(x_t) - g(x_{t+1}) \geq \nabla f(x_t)^\top (x_{t+1} - x_t) + \frac{p_b}{\eta} D_\phi(x_{t+1}, x_t). \tag{19}$$

Note that this crucially relies on $x_{t+1} - x_t$ having support on $b$, which is enforced by the block-separability of $\phi$. Then, the proof is similar to that of the first point, using that $\eta_t L_{\text{rel}}^b \leq p_b$. $\qquad \square$

Using this monotonicity result allows us to prove Theorem 3.

*Proof of Theorem 3.* First note that by convexity of all $g_i$,

$$\nabla^2 V_t^b(x) = \sum_{i \in b} \frac{\eta_t}{p_i} \nabla^2 g_i(x^{(i)}) + \nabla^2 \phi(x) \succcurlyeq \nabla^2 \phi(x).$$

Therefore, we have $D_{V_t^b}(x, y) \geq D_\phi(x, y)$ for all $x, y \in \mathbb{R}^d$. Applying this with $y = x_{t+1}$ yields:

$$V_t^b(x) - V_t^b(x_{t+1}) - \nabla V_t^b(x_{t+1})^\top (x - x_{t+1}) \geq D_\phi(x, x_{t+1}). \tag{20}$$

Then, $\nabla V_t^b(x_{t+1}) = 0$ by definition of $x_{t+1}$, so Equation (20) writes:

$$D_\phi(x, x_{t+1}) + \sum_{i \in b} \frac{\eta_t}{p_i} \left( g_i(x_{t+1}^{(i)}) - g(x^{(i)}) \right) \leq \sum_{i \in b} \frac{\eta_t}{p_i} \nabla_i f(x_t)^\top (x - x_{t+1})$$
$$+ D_\phi(x, x_t) - D_\phi(x_{t+1}, x_t).$$

We first consider that the first option of Assumption 1 holds, *i.e.*, that $f$ and $\phi$ are separable in $b$. We note $\delta_i = e_i^\top (x_{t+1} - x_t) e_i$, so that:

$$-\nabla_i f(x_t)^\top (x_{t+1} - x_t) = \nabla f(x_t)^\top (x_t + \delta_i - x_t)$$
$$= f(x_t) - f(x_t + \delta_i) + D_f(x_t + \delta_i, x_t)$$
$$\leq f(x_t) - f(x_t + \delta_i) + L_{\text{rel}}^i D_{\phi_i}(x_{t+1}^{(i)}, x_t^{(i)}).$$

Therefore, if $\eta_t L_{\text{rel}}^i \leq p_i$ for all $i \in b$,

$$-\sum_{i \in b} \frac{\eta_t}{p_i} \nabla_i f(x_t)^\top (x_{t+1} - x_t) - D_\phi(x_{t+1}, x_t)$$

$$\leq \sum_{i \in b} \frac{\eta_t}{p_i} [f(x_t) - f(x_t + \delta_i)] + \sum_{i \in b} \left( \frac{\eta_t L_{\text{rel}}^i}{p_i} - 1 \right) D_{\phi_i}(x_{t+1}^{(i)}, x_t^{(i)})$$

$$\leq \sum_{i \in b} \frac{\eta_t}{p_i} [f(x_t) - f(x_t + \delta_i)]$$

The $g_i(x_{t+1}^{(i)}) - g_i(x^{(i)})$ term can be replaced by $g(x_t + \delta_i) - g(x_t) + g_i(x_t^{(i)}) - g_i(x^{(i)})$ since $g_j(x_{t+1}) = g_j(x_t)$ for $j \neq i$. Therefore, we obtain:

$$D_\phi(x, x_{t+1}) + \sum_{i \in b} \frac{\eta_t}{p_i} [F(x_t + \delta_i) - F(x_t)] + \sum_{i \in b} \frac{\eta_t}{p_i} \left( g_i(x_t^{(i)}) - g_i(x^{(i)}) \right)$$
$$\leq \sum_{i \in b} \frac{\eta_t}{p_i} \nabla_i f(x_t)^\top (x - x_t) + D_\phi(x, x_t). \tag{21}$$

The separability of $F$ in $b$ and its monotonicity lead to, using the fact that $x_{t+1} = x_t + \sum_{i \in b} \delta_i$:

$$\sum_{i \in b} \frac{\eta_t}{p_i} [F(x_t + \delta_i) - F(x_t)] \geq \frac{\eta_t}{p_{\min}} \sum_{i \in b} [F(x_t + \delta_i) - F(x_t)] = \frac{\eta_t}{p_{\min}} [F(x_{t+1}) - F(x_t)].$$

Therefore, if the first option of Assumption 1 holds, we obtain:

$$D_\phi(x, x_{t+1}) + \frac{\eta_t}{p_{\min}} [F(x_{t+1}) - F(x_t)] + \sum_{i \in b} \frac{\eta_t}{p_i} \left( g_i(x_t^{(i)}) - g_i(x^{(i)}) \right)$$
$$\leq \sum_{i \in b} \frac{\eta_t}{p_i} \nabla_i f(x_t)^\top (x - x_t) + D_\phi(x, x_t). \tag{22}$$

If the second option holds, *i.e.*, $p_i = p$ for all $i \in b$, then

$$\sum_{i \in b} \frac{\eta_t}{p_i} \nabla_i f(x_t)^\top (x_{t+1} - x_t) = \frac{\eta_t}{p} \nabla f(x_t)^\top (x_{t+1} - x_t),$$

and Equation (22) can be obtained through similar derivations (at the block-level). Using the separability of $g$, we obtain that

$$\mathbb{E}\left[ \sum_{i \in b} \frac{1}{p_i} \left( g_i(x_t^{(i)}) - g_i(x^{(i)}) \right) \right] = g(x_t) - g(x).$$

Then, since $\mathbb{E}\left[\sum_{i\in b}\frac{1}{p_i}\nabla_i f(x_t)\right] = \sum_i p_i^{-1}\sum_{b:i\in b} p(b)\nabla_i f(x_t) = \nabla f(x_t)$, and the relative strong convexity assumption yields:

$$\mathbb{E}\left[\sum_{i\in b}\frac{1}{p_i}\nabla_i f(x_t)^\top (x - x_t)\right] = \nabla f(x_t)^\top (x - x_t) \le f(x) - f(x_t) - \sigma_{\mathrm{rel}}D_\phi(x, x_t).$$

Therefore, taking the expectation of Equation (21) yields:

$$\mathbb{E}\left[D_\phi(x, x_{t+1}) + \frac{\eta_t}{p_{\min}}\left(F(x_{t+1}) - F(x_t)\right)\right] \le \eta_t\left(F(x) - F(x_t)\right) + (1 - \eta_t\sigma_{\mathrm{rel}})D_\phi(x, x_t).$$

We obtain after some rewriting:

$$\mathbb{E}\left[D_\phi(x, x_{t+1}) + \frac{\eta_t}{p_{\min}}\left(F(x_{t+1}) - F(x)\right)\right]$$
$$\le (1 - p_{\min})\frac{\eta_t}{p_{\min}}\left(F(x_t) - F(x)\right) + (1 - \eta_t\sigma_{\mathrm{rel}})D_\phi(x, x_t).$$

Finally, $\sigma_{\mathrm{rel}} \le L_{\mathrm{rel}}^i$ so $\eta_t\sigma_{\mathrm{rel}} \le \eta_t L_{\mathrm{rel}}^i \le p_i$ for all $i$, and in particular $1 - p_{\min} \le 1 - \eta_t\sigma_{\mathrm{rel}}$, which yields the desired result.

The result on $L_t'$ is be obtained by bounding $\eta/p_{\min}$ by $L_{\mathrm{rel}}^{\max} = \max_i L_{\mathrm{rel}}^i$ and remarking that $1 - \eta_t L_{\mathrm{rel}}^{\max} \le 1 - \eta_t\sigma_{\mathrm{rel}}$ since $L_{\mathrm{rel}}^{\max} \ge \sigma_{\mathrm{rel}}$. $\qquad\square$

# B  Convergence results for DVR

We now give a series of small results, that justify our approach. We start by showing the applicability of Theorem 3 to Problem (4), and the associated constants. Finally, we show how to obtain rates for the primal iterates $\theta_t$.

## B.1  Application to the dual of the augmented problem

In this section, we note $f_{\mathrm{sum}}^* = \sum_{i=1}^n\sum_{j=1}^m f_{ij}^*$, so that Problem 4 writes:

$$\min_{x,y} q_A(x, y) + f_{\mathrm{sum}}^*(y) \tag{23}$$

**Lemma 2.** *The iterations of Algorithm 1 are equivalent to the iteration of Equations (14) applied to Problem (4) with $g = 0$ and $\phi(x, y) = \phi_{\mathrm{comm}}(x) + \sum_{i=1}^n\sum_{j=1}^m \phi_{ij}(y^{(ij)})$, with $\phi_{\mathrm{comm}}(x) = \frac{1}{2}\|x\|_{A^\dagger A}^2$ for coordinates associated with communication edges, and $\phi_{ij}(y^{(ij)}) = \frac{L_{ij}}{\mu_{ij}^2}f_{ij}^*(\mu_{ij}y_{ij})$ for coordinates associated with computation edges.*

*Proof.* This result follows from the dual-free and implementation-friendly derivations presented in the previous section. $\qquad\square$

**Lemma 3.** *Let $\alpha = 2\lambda_{\min}(A_{\mathrm{comm}}^\top D_M^{-1} A_{\mathrm{comm}})$, and $\phi$ as in Lemma 2, then:*

1. *$q_A + f_{\mathrm{sum}}^*$ is $(\alpha/2)$-strongly convex relatively to $\phi$.*

2. *$q_A + f_{\mathrm{sum}}^*$ is $(L_{\mathrm{rel}}^{\mathrm{comm}})$-smooth relatively to $\phi$ in the direction of communication edges, with*

$$L_{\mathrm{rel}}^{\mathrm{comm}} = \lambda_{\max}(A_{\mathrm{comm}}^\top \Sigma_{\mathrm{comm}} A_{\mathrm{comm}}).$$

3. *$q_A + f_{\mathrm{sum}}^*$ is $(L_{\mathrm{rel}}^{ij})$-smooth relatively to $\phi$ in the direction of virtual edge $(i, j)$, with*

$$L_{\mathrm{rel}}^{ij} = \alpha\left(1 + \frac{L_{ij}}{\sigma_i}\right).$$

*Proof.* First note that $\nabla^2 f_{\mathrm{sum}}^*$ is a block-diagonal matrix, and its $ij$-th block is equal to

$$(\nabla^2 f_{\mathrm{sum}}^*(y))_{ij} = A^\top(u_{ij}u_{ij}^\top \otimes \nabla^2 f_{ij}^*(\mu_{ij}y^{(ij)}))A \succcurlyeq \frac{1}{L_{ij}}A^\top(u_{ij}u_{ij}^\top \otimes P_{ij})A, \tag{24}$$

where $u_{ij} \in \mathbb{R}^{n(1+m)}$ denotes the unit vector corresponding to virtual *node* $(i,j)$. We denote $\tilde{\Sigma} = \Sigma + \sum_{i=1}^{n} \sum_{j=1}^{m} \frac{1}{L_{ij}} (u_{ij} u_{ij}^\top) \otimes I_d$. Then,

$$\nabla^2 q_A(x,y) + \nabla^2 f_{\text{sum}}^*(y) = A^\top \tilde{\Sigma} A + \nabla^2 f_{\text{sum}}^*(y) - A^\top \left[ \sum_{i=1}^{n} \sum_{j=1}^{m} \frac{1}{L_{ij}} (u_{ij} u_{ij}^\top) \otimes P_{ij} \right] A. \quad (25)$$

**Relative strong convexity.** Then, [13, Lemma 6.5] leads to $A^\top \tilde{\Sigma} A \succcurlyeq \sigma_F A^\dagger A$. Note that the notations are slightly different, and the matrix $\tilde{\Sigma}$ in this paper is the same as the matrix $\Sigma^\dagger$ in [13]. Then, remark that $(A^\dagger A)_{ij} = P_{ij} = \frac{1}{\mu_{ij}^2} (A^\top [(u_{ij} u_{ij}^\top) \otimes P_{ij}] A)_{ij}$, and $\phi_{ij} = \alpha^{-1} f_{ij}^*$, so that:

$$\nabla^2 q_A(x,y) + \nabla^2 f_{\text{sum}}^*(y) \succcurlyeq \sigma_F \nabla^2 \phi(x,y) +$$
$$(1 - \alpha^{-1}\sigma_F) \left[ \nabla^2 f_{\text{sum}}^*(y) - A^\top \left[ \sum_{i=1}^{n} \sum_{j=1}^{m} \frac{1}{L_{ij}} (u_{ij} u_{ij}^\top) \otimes P_{ij} \right] A. \right].$$

Finally, using that Equation (24) along with the fact that $\sigma_F \leq \alpha$ implies that $q_A + f_{\text{sum}}^*$ is $\sigma_F$-relatively strongly convex with respect to $\phi$.

**Relative smoothness.** We first prove the relative smoothness property for communicate edges. For any $\tilde{x} \in R^{Ed}$, Equation (25) leads to:

$$(\tilde{x}, 0)^\top [\nabla^2 q_A(x,y) + \nabla^2 f_{\text{sum}}^*(y)](\tilde{x}, 0) = (\tilde{x}, 0)^\top A^\top \Sigma A (\tilde{x}, 0) \preccurlyeq L_{\text{rel}}^{\text{comm}} (\tilde{x}, 0)^\top \nabla^2 \phi(x,y)(\tilde{x}, 0).$$

Similarly, for any $\theta \in \mathbb{R}^d$, we consider $\tilde{y} = e_{ij} \otimes \theta$ and write:

$$\tilde{y}^\top [\nabla^2 q_A(x,y) + \nabla^2 f_{\text{sum}}^*(y)] \tilde{y} = \tilde{y}^\top A^\top \tilde{\Sigma} A \tilde{y} + \mu_{ij}^2 \theta^\top \left[ \nabla^2 f_{ij}^*(\mu_{ij} y^{(ij)}) - \frac{1}{L_{ij}} P_{ij} \right] \theta$$
$$\preccurlyeq L_{\text{rel}}^i \tilde{y}^\top \nabla^2 \phi(x,y) \tilde{y} + (1 - \alpha^{-1} L_{\text{rel}}^i) \theta^\top \left[ \nabla^2 f_{ij}^*(\mu_{ij} y^{(ij)}) - \frac{1}{L_{ij}} P_{ij} \right] \theta,$$

with

$$L_{\text{rel}}^i = \max_\theta \mu_{ij}^2 u_{ij}^\top \tilde{\Sigma} u_{ij} \frac{\theta^\top P_{ij} \theta}{\|\theta\|^2} \leq \alpha \left( 1 + \frac{L_{ij}}{\sigma_i} \right).$$

Finally, $\nabla^2 f_{ij}^*(\mu_{ij} y^{(ij)}) \succcurlyeq P_{ij}/L_{ij}$, and $\alpha \leq L_{\text{rel}}^i$, which ends the proof of the directional relative smoothness result. $\square$

**Lemma 4.** *Assumption 1 holds with $f = q_A + f_{\text{sum}}^*$, $g = 0$, and $\phi$ as in Lemma 2, and when the sampling is such that either:*

- *All communication edges are sampled at once, or*
- *Each node samples exactly one virtual edge.*

*Proof.* First of all, $g = 0$ is separable, and $\phi$ is separable with respect to the communication and computation blocks by construction.

We note $b_{\text{comm}}$ the block of all communication edges, which is sampled with probability $p_{\text{comm}}$. All communication edges are sampled at the same time, so $p_i = p_{\text{comm}}$ for all $i \in b_{\text{comm}}$ and so $\phi$ respects option 2 for the communication block.

Let us now consider a computation block $b$. First of all, $\phi$ is separable for the virtual edges. Then, virtual blocks contain exactly one virtual edge per node, and so $b = \{(1, j_1), \cdots, (n, j_n)\}$. Let $k \neq \ell$, then

$$e_{k,j_k}^\top A^\top \Sigma A e_{\ell, j_\ell} = \mu_{k,j_k} \mu_{\ell,j_\ell} (e_k - e_{k,j_k})^\top \Sigma (e_\ell - e_{\ell,j_\ell}) = 0.$$

Therefore,

$$q_A\left(x_t + \sum_{(i,j)\in b} \delta_{ij}\right) - q_A(x_t) = \frac{1}{2}\left(\sum_{(i,j)\in b} \delta_{ij}\right) A^\top \Sigma A\left(\sum_{(ij)\in b} \delta_{ij}\right) + \left(\sum_{(ij)\in b} A\delta_{ij}\right)^\top \Sigma A x_t$$

$$= \sum_{(i,j)\in b} \left(q_A(\delta_{ij}) + \delta_{ij}^\top A^\top \Sigma A x_t\right)$$

$$= \sum_{(i,j)\in b} \left(q_A(x_t + \delta_{ij}) - q_A(x_t)\right).$$

Finally, $f_{\text{sum}}^*$ is separable, and so $q_A + f_{\text{sum}}^*$ respects option 2. $\square$

We can now prove the main theorem on the convergence rate of DVR.

**Theorem 4.** *We choose* $p_{\text{comm}} = \left(1 + \gamma \frac{m+\kappa_s}{\kappa_{\text{comm}}}\right)^{-1}$ *and* $p_{ij} \propto (1 - p_{\text{comm}})(1 + L_{ij}/\sigma_i)$. *Then, for all* $\theta_0 \in \mathbb{R}^{n\times d}$ *and all* $t > 0$, *the error is such that:*

$$\frac{\eta_t}{p_{\min}} D_\phi(\lambda_\star, \lambda_t) + D(\lambda_t) - D(\lambda_\star) \leq \left(1 - \frac{\alpha\eta_t}{2}\right)^t \left[\frac{\eta_t}{p_{\min}} D_\phi(\lambda_\star, \lambda_0) + D(\lambda_0) - D(\lambda_\star)\right], \quad (26)$$

*with* $p_{\min} = \min(p_{\text{comm}}, \min_{ij} p_{ij})$, $\lambda_t = (x_t, y_t)$ *and* $D = -(q_A + f_{\text{sum}}^*)$. *Therefore, the expected time* $T_\varepsilon$ *required to reach precision* $\varepsilon$ *is equal to:*

$$T_\varepsilon = O\left(\left[2(m + \kappa_s) + \tau\frac{\kappa_{\text{comm}}}{\gamma}\right]\log\varepsilon^{-1}\right).$$

*Proof.* Using Lemmas 4 and 3, we apply Theorem 3 (convergence of Bregman coordinate gradient descent), and obtain that the convergence rate is $\eta_t\alpha/2$, with $\eta_t \leq \min_{ij} p_{ij}/L_{\text{rel}}^i$ and $\eta_t \leq p_{\text{comm}}/L_{\text{rel}}^{\text{comm}}$. Therefore, for communication edges, we have that

$$\eta_t \leq \frac{p_{\text{comm}}}{L_{\text{rel}}^{\text{comm}}} = \frac{p_{\text{comm}}}{\lambda_{\max}(A_{\text{comm}}^\top \Sigma_{\text{comm}}^{-1} A_{\text{comm}})}.$$

For computation edges, we know that $p_{ij} = p_{\text{comm}}(1 + L_{ij}/\sigma_i)/(\sum_{j=1}^m (1 + L_{ij}/\sigma_i))$, and so

$$\eta_t \leq \frac{p_{ij}}{L_{\text{rel}}^{ij}} = \frac{p_{\text{comp}}}{\alpha\sum_{j=1}^m (1 + \sigma_i^{-1}L_{ij})} \leq \frac{p_{\text{comp}}}{\alpha(m + \kappa_s)},$$

with $\kappa_s \geq \sigma_i^{-1}\sum_{j=1}^m L_{ij}$ for all $i$.

In the end, we would like these two bounds to be equal, so we choose $p_{\text{comp}}$ and $p_{\text{comm}}$ such that

$$p_{\text{comp}} = p_{\text{comm}}(m + \kappa_s)\frac{\lambda_{\min}^+(A_{\text{comm}}^\top D_M^{-1} A_{\text{comm}})}{\lambda_{\max}(A_{\text{comm}}^\top \Sigma_{\text{comm}}^{-1} A_{\text{comm}})}.$$

Yet, we also know that $p_{\text{comm}} = 1 - p_{\text{comp}}$, so

$$p_{\text{comp}} = \left(1 + \frac{1}{m + \kappa_s}\frac{\lambda_{\max}(A_{\text{comm}}^\top \Sigma_{\text{comm}}^{-1} A_{\text{comm}})}{\lambda_{\min}^+(A_{\text{comm}}^\top D_M^{-1} A_{\text{comm}})}\right)^{-1}.$$

Equivalently, this corresponds to taking

$$p_{\text{comm}} = \left(1 + \gamma\frac{m + \kappa_s}{\kappa_{\text{comm}}}\right)^{-1}.$$

With this choice, one can verify that $\eta_t$ verifies both $\eta_t\alpha \leq 2p_{\text{comm}}$ and $\eta_t\alpha \leq 2\min_{ij} p_{ij}$, so the rate is:

$$1 - \frac{\eta_t\alpha}{2} = 1 - \frac{p_{\text{comp}}}{2(m + \kappa_s)}.$$

The expected execution time to reach precision $\varepsilon$, denoted $T_\varepsilon$, is equal to $T_\varepsilon = \rho^{-1}(p_{\text{comp}} + \tau p_{\text{comm}})K_\varepsilon$ with $K_\varepsilon$ such that $C(1 - \eta_t\alpha/2)^{K_\varepsilon} < \varepsilon$ for some constant $C$, and so:

$$T_\varepsilon = O\left(2(m + \kappa_s) + \tau\frac{\kappa_{\text{comm}}}{\gamma}\right).$$

$\square$

## B.2 Primal guarantees

The goal of this section is to recover primal guarantees from dual guarantees. Although the initial setting is inspired from [24], the proof is different, and in particular does not require smoothness of the $f_{ij}^*$ or an extra proximal step. We define for $\beta \geq 0$ the Lagrangian function:

$$\mathcal{L}(\lambda, \theta) = \sum_{i=1}^{n} \sum_{j=1}^{m} f_{ij}(\theta^{(ij)}) + \frac{\sigma_i}{2}\|\theta^{(i)}\|^2 + \frac{\beta}{2}\|\theta^{(i)} - \omega^{(i)}\|^2 - \lambda^\top A^\top \theta. \tag{27}$$

The dual problem $D(\lambda)$ is defined as

$$D(\lambda) = \min_{\theta} \mathcal{L}(\lambda, \theta).$$

Given an approximate dual solution $\lambda_k$, we can get an approximate primal solution $\theta_k = \arg\min_\theta \mathcal{L}(\lambda_k, \theta)$, which is obtained as:

$$\theta_t^{(ij)} = \arg\min_v \left( f_{ij}(v) - \mu_{ij}\lambda_t^{(ij)}v \right) \in \partial f_{ij}^*(\mu_{ij}\lambda_t^{(ij)}), \tag{28}$$

$$\theta_t^{(i)} = \frac{1}{\sigma_i + \beta}\left( (A\lambda_t)^{(i)} + \beta\omega^{(i)} \right). \tag{29}$$

Note that $\theta_t^{(ij)}$ corresponds to the $z_t^{(ij)}$ from Algorithm 1. We chose to use a different notation in the main text to emphasize on the fact that these are the parameters for the virtual nodes, but $z_t^{(ij)}$ actually converge to the solution as well. Similarly, $\lambda_t$ corresponds to $(x_t, y_t)$ the concatenation of the parameters for communication and virtual edges from Section 2. The last difference is that the Lagrangian defined in Equation (27) actually corresponds to a Lagrangian associated to a perturbed version of Problem (1) in which $\tilde{f}_i(\theta) = f_i(\theta) + \frac{\beta}{2}\|\theta - \omega^{(i)}\|^2$. The solution to the initial problem can be retrieved by taking $\beta = 0$, but this more general formulation enables us to derive results that also holds for the inner problems solved by the Catalyst accelerated version of DVR.

**Lemma 5.** *Denote* $C_0 = \frac{(\beta + \sigma_{\max} + L_{\max})}{2(\sigma_{\min} + \beta)^2}\left( \frac{p_{\min}}{\eta_t}D_\phi(\lambda^\star, \lambda_0) + (D(\lambda^\star) - D(\lambda_0)) \right)$, *then*

$$\sum_{i=1}^{n} \|\theta_t^{(i)} - \theta^\star\|^2 \leq C_0(1 - \rho)^t. \tag{30}$$

*Proof.* Using the fact that $\theta_t^{(i)} = \frac{1}{\sigma_i + \beta}((A\lambda_t)^{(i)} + \omega_t^{(i)})$ (and similarly for $\theta^\star$), where $\Sigma_\beta$ is the block diagonal matrix such that $(\Sigma_\beta)_{ii} = (\sigma_i + \beta)^{-1}I_d$, we obtain:

$$\sum_{i=1}^{n} \|\theta_t^{(i)} - \theta^\star\|^2 = \sum_{i=1}^{n} \frac{1}{(\sigma_i + \beta)^2}\|(A\lambda_t)^{(i)} - (A\lambda^\star)^{(i)}\|^2$$

$$\leq \frac{1}{(\sigma_{\min} + \beta)^2}\|A\lambda_t - A\lambda^\star\|^2.$$

Using the $\min((\sigma_{\max} + \beta)^{-1}, L_{ij}^{-1})$-strong convexity of $\theta \mapsto \frac{1}{2}x^\top\Sigma_\beta x + \sum_{i,j} f_{ij}^*(x^{(ij)})$, we obtain:

$$\sum_{i=1}^{n} \|\theta_t^{(i)} - \theta^\star\|^2 \leq 2\frac{(\beta + \sigma_{\max} + L_{\max})}{(\sigma_{\min} + \beta)^2}\left(D(\lambda^\star) - D(\lambda_t)\right). \tag{31}$$

Then, we add $p_{\min}\eta_t^{-1}D_\phi(\lambda^\star, \lambda_t) \geq 0$ and apply Theorem 4, which yields

$$\sum_{i=1}^{n} \|\theta_t^{(i)} - \theta^\star\|^2 \leq \frac{2(\beta + \sigma_{\max} + L_{\max})}{(\sigma_{\min} + \beta)^2}(1 - \rho)^t\left(\frac{p_{\min}}{\eta_t}D_\phi(\lambda^\star, \lambda_0) + (D(\lambda^\star) - D(\lambda_0))\right).$$

$\square$

Then, Theorem 1 is a direct consequence of Theorem 4 and Lemma 5.

## C  Catalyst acceleration

We show in this Section how to apply Catalyst acceleration to DVR, and prove the convergence speed in this case.

### C.1  Derivation and rates

In the main text, we derived DVR to solve regularized finite sum problems. Although not so different, the subproblem obtained with Catalyst is not in the form of Problem (1), and some adjustments need to be made. More specifically, we would like to solve problems of the form:

$$\min_\theta \left\{ F_t(\theta) \triangleq \sum_{i=1}^n \left[ \frac{\sigma_i}{2} \|\theta\|^2 + \frac{\beta}{2} \|\theta - \omega_t^{(i)}\|^2 + \sum_{j=1}^m f_{ij}(\theta) \right] \right\}. \tag{32}$$

An easy way to adapt the algorithm is to consider the extra $(\beta/2)\|\theta - \omega_t^{(i)}\|^2$ as just another component of the sum. Yet, the point of this extra term is to make the problem easier to solve by adding strong convexity. This would not be the case if this term were is treated as just another term in the sum. Therefore, we want to include it with the quadratic term. We define:

$$h(x) = \frac{\sigma_i}{2} \|\theta\|^2 + \frac{\beta}{2} \|\theta - \omega_t^{(i)}\|^2,$$

then $h^*(x) = \frac{1}{2(\beta+\sigma)} \|x + \beta\omega_t^{(i)}\|^2 - \frac{\beta}{2} \|\omega_t^{(i)}\|^2$. Therefore, Problem (4) becomes:

$$\min_{\lambda \in \mathbb{R}^{(E+mn)d}} \frac{1}{2} \lambda^\top A^\top \Sigma_\beta A \lambda + \beta \omega_t^\top \Sigma_\beta A \lambda + \sum_{i=1}^n \sum_{j=1}^m f_{ij}^*((A\lambda)_{ij}), \tag{33}$$

with $(\Sigma_\beta)_{ii} = (\sigma_i + \beta)^{-1}$ for $i \in \{1, \dots, n\}$. The linear term does not affect the Hessians, and thus the convergence rate is the same as before, with $\sigma$ replaced by $\sigma + \beta$. In terms of algorithms, we just need to modify the gradient term, and obtain Algorithm 2. The only term that changes is $\nabla q_A(x, y)$, to which an extra $\beta \Sigma_\beta \omega_t$ term is added. Therefore, the updates to $\theta_t$ and $z_t$ remain unchanged, and only the initial expression of $\theta_t$ requires some adjustments since we now have that (as written in Equation 29):

$$\theta_{t,k}^{(i)} = \frac{1}{\sigma_i + \beta} \left( (A\lambda_{t,k})^{(i)} + \beta\omega^{(i)} \right).$$

If we only consider 1 inner loop then the only thing that changes is the initial condition. If we consider several outer loops, then the we must choose the new parameter as $\theta_0^{t+1} = \theta_T^t + \Sigma_\beta (\omega_{t+1} - \omega_t)$ in order to maintain the invariant, but a remarkable fact is that the inner iterations remain the same, with the only exception that $\Sigma$ is replaced by $\Sigma_\beta$. Note that it is possible to warm-start the $z_{t+1,0}$ as well, but this requires updating $\theta_{t,0}$ accordingly with $\nabla f_{ij}(z_{t,0}^{(ij)})$, which requires a full pass over the local dataset. We therefore choose not to do it.

However, it is not obvious that Algorithm 2 corresponds to a genuine Catalyst acceleration yet. Indeed, Catalyst acceleration requires having a feasible $\varepsilon_t$-approximations for the primal problem, *i.e.*, points $\theta_t \in \mathbb{R}^d$ such that $F_t(\theta_t) - \min_\theta F(\theta) \leq \varepsilon_t$. In our case, we only have dual guarantees and approximate feasibility. We know that the parameters converge to consensus, but they do not reach it at any time. This is a problem because it is then not possible to adequately define $F_{t+1}$ based on the local approximations of the solutions of $F_t$. Yet, following the approach of [20], we note that

$$\sum_{i=1}^n \|\theta - \omega_t^{(i)}\|^2 = n\|\theta - \bar{\omega}_t\|^2 + \sum_{i=1}^n \|\omega_t^{(i)}\|^2 - n\|\bar{\omega}_t\|^2,$$

where $\bar{\omega}_t = \frac{1}{n} \sum_{i=1}^n \omega_t^{(i)}$. This means that although $F_t$ is only defined with the local variables $\omega_t^{(i)}$, *solving $F_t$ is equivalent to solving a problem involving $\bar{\omega}_t$ only*. Besides, the Catalyst iterations are linear, meaning that performing the extrapolation step on $\bar{\theta}_t$ is equivalent to performing it on each $\theta_t^{(i)}$ individually. Therefore, although Catalyst is implemented in a fully decentralized manner (each node

**Algorithm 2** Accelerated DVR($z_0$)

---

1: $\alpha = 2\lambda_{\min}^+(A_{\text{comm}}^\top D_M^{-1} A_{\text{comm}})$, $\eta = \min\left(\frac{p_{\text{comm}}}{\lambda_{\max}(A_{\text{comm}}^\top \Sigma_{\beta,\text{comm}} A_{\text{comm}})}, \frac{p_{ij}}{\alpha(1+\sigma_i^{-1}L_{ij})}\right)$

2: $q = \frac{\sigma_{\min}}{\sigma_{\min}+\beta}$       *// Initialization*

3: $\omega_0^{(i)} = -\frac{1}{\sigma_i+\beta}\sum_{j=1}^m \nabla f_{ij}(z_0^{(ij)})$, $\theta_0^{(i)} = \left(1+\frac{\beta}{\sigma_i+\beta}\right)\omega_0^{(i)}$.       *// $z_0$ is arbitrary but not $\theta_0$.*

4: **for** $t = 0$ to $T-1$ **do**       *// T outer loops*

5:     **for** $k = 0$ to $K-1$ **do**       *// Inner loop runs for K iterations*

6:         $z_{t,k+1} = z_{t,k}$.

7:         Sample $u_t$ uniformly in $[0,1]$.       *// Randomly decide the kind of update*

8:         **if** $u_t \le p_{\text{comm}}$ **then**

9:             $\theta_{t,k+1} = \theta_{t,k} - \frac{\eta_t}{p_{\text{comm}}}\Sigma_\beta W \theta_{t,k}$       *// Communication using W*

10:         **else**

11:             **for** $i = 1$ to $n$ **do**

12:                 Sample $j \in \{1, \cdots, m\}$ with probability $p_{ij}$.

13:                 $z_{t,k+1}^{(ij)} = \left(1-\frac{\alpha\eta}{p_{\text{comp}}}\right)z_{t,k}^{(ij)} + \frac{\alpha\eta}{p_{\text{comp}}}\theta_{t,k}^{(i)}$       *// Computing new virtual node parameter*

14:                 $\theta_{t,k+1}^{(i)} = \theta_{t,k}^{(i)} - \frac{1}{\sigma_i+\beta}\left(\nabla f_{ij}(z_{t,k+1}^{(ij)}) - \nabla f_{ij}(z_{t,k}^{(ij)})\right)$       *// Local update using $f_{ij}$*

15:     $\omega_{t+1} = \theta_{t,K} + \frac{1-\sqrt{q}}{1+\sqrt{q}}(\theta_{t,K} - \theta_{t-1,K})$

16:     $\theta_{t+1,0} = \theta_{t,K} + \frac{\beta}{\beta+\sigma_i}(\omega_{t+1} - \omega_t)$

17:     $z_{t+1,0} = z_{t,K}$

18: **return** $\theta_T$

---

knowing only its own parameter), it is conceptually applied to a mean parameter $\bar{\theta}_t$ (that is never explicitly computed). In the following, we thus analyze the performances of the following algorithm:

$$\bar{\theta}_{t+1} \approx \arg\min_\theta F(\theta) + \frac{n\beta}{2}\|\theta - \bar{\omega}_t\|^2$$
$$\bar{\omega}_t = \bar{\theta}_{t+1} + \frac{1-\sqrt{q}}{1+\sqrt{q}}(\bar{\theta}_{t+1} - \bar{\theta}_t), \tag{34}$$

where we recall that $q = \sigma_{\min}/(\sigma_{\min}+\beta)$. Recall that the inner problem is approximated using DVR and the means do not need to be computed explicitly. Let $\kappa_s^\beta = \max_i 1 + (\sum_{j=1}^m L_{ij})/(\beta+\sigma_i)$, and $\kappa_{\text{comm}}^\beta$ be obtained similarly to $\kappa_{\text{comm}}$ but replacing $\Sigma$ by $\Sigma_\beta$. We consider in this section that $\sigma_i = \sigma$ for all $i \in \{1,\ldots,n\}$ in order to simplify exposition, but the results hold more generally. Note that $\alpha$ and $\eta$ have slightly different expressions than in the main text since $\beta$ is now involved in their definitions. We define the sequence $\varepsilon_t$ which is such that:

$$\varepsilon_t = \frac{2}{9}\left(F(\theta_0) - F(\theta^\star)\right)\left(1-\rho^{\text{out}}\right)^t \text{ with } \rho^{\text{out}} < \sqrt{q}, \text{ and } q = \frac{\sigma}{\sigma+\beta}. \tag{35}$$

We then prove the following theorem:

**Theorem 5.** *Consider Algorithm 2 with $p_{\text{comm}} = \left(1+\gamma\frac{m+\kappa_s^\beta}{\kappa_{\text{comm}}^\beta}\right)^{-1}$, $p_{ij} \propto (1-p_{\text{comm}})(1+L_{ij}/(\sigma_i+\beta))$. If $K = \tilde{O}\left(1/(\eta_t\alpha)\right)$ then for all $t \le T$, $F_t(\bar{\theta}_t) - F_t(\theta_t^\star) \le \varepsilon_t$ and*

$$F(\bar{\theta}_t) - F(\theta^\star) \le \frac{8}{(\sqrt{q}-\rho^{\text{out}})^2}(1-\rho^{\text{out}})^{t+1}(F(\bar{\theta}_0) - F(\theta^\star)). \tag{36}$$

Note that the error is on the mean parameter, and we also want $\theta_t^{(i)}$ to be close to $\bar{\theta}_t$ for all $i$. This is ensured by Lemma 5. Before we start the proof of Theorem 5, we show that Theorem 2 is a corollary of Theorem 5.

*Proof of Theorem 2.* Using the same argument as in Theorem 1, we obtain that each inner loop takes time

$$T_{\text{inner}} = O\left(m + \frac{L_s+\sigma}{\beta+\sigma} + \tau\frac{L_{\text{comm}}+\beta}{\gamma(\beta+\sigma)}\right)$$

in expectation, so the total number of inner iterations is of order:

$$T_\varepsilon = \tilde{O}\left(\sum_{k=0}^{\lceil 1/\rho^{\text{out}}\rceil} T_{\text{inner}}\right) = \tilde{O}\left(\sqrt{1 + \frac{\beta}{\sigma}}\left(m + \frac{L_s + \sigma}{\beta + \sigma} + \tau\frac{L_{\text{comm}} + \beta}{\gamma(\beta + \sigma)}\right)\log\frac{1}{\varepsilon}\right). \qquad (37)$$

Therefore, we see that if we choose $\beta + \sigma = L_{\text{comm}}$ then, taking into account the fact that $\kappa_s \leq m\kappa_{\text{comm}}$, the algorithm takes time:

$$T_\varepsilon = \tilde{O}\left(\sqrt{\kappa_{\text{comm}}}\left(m + \frac{\tau_c}{\gamma}\right)\right).$$

Therefore, using Chebyshev acceleration allows to recover the rate of optimal batch algorithms (up to log factors). On the other hand, if we choose $\beta = L_s/m - \sigma$ then if $\beta \geq 0$ (*i.e.*, $\kappa_s \geq m$), the time to convergence is equal to:

$$T_\varepsilon = \tilde{O}\left(\sqrt{\frac{\kappa_s}{m}}\left(m + \tau\frac{m\kappa_{\text{comm}} + \kappa_s}{\gamma\kappa_s}\right)\right).$$

This can be rewritten as:

$$T_\varepsilon = \tilde{O}\left(\sqrt{m\kappa_s} + \tau\frac{\sqrt{\kappa_{\text{comm}}}}{\gamma}\sqrt{\frac{m\kappa_{\text{comm}}}{\kappa_s}}\right).$$

Therefore, we obtain the optimal $\sqrt{m\kappa_s}$ computation complexity in this case, with a slightly suboptimal communication complexity due to the $\sqrt{m\kappa_{\text{comm}}/\kappa_s}$ term. When this term is equal to $1$ then $\sqrt{m\kappa_s} = m\sqrt{\kappa_b}$ and so nothing is gained from using a stochastic algorithm. Otherwise, this allows to trade-off communications for computations. $\qquad\square$

The proof of Theorem 5 is obtained in several steps, that we emphasize below:

1. Equivalent decentralized implementation of Catalyst.
2. Bounding the primal suboptimality as $F_t(\bar{\theta}_t) - \min_\theta F_t(\theta) \leq (1 - (\eta\alpha)/2)^k D_0^t$, with $k$ the number of inner iterations and $D_0^t$ a dual error. This quantifies how precisely the inner problem is solved.
3. Evaluating the initial dual suboptimality $D_0^t$, which depends on $\theta_{t-1}$ (and its associated dual parameter $\lambda_{t-1}$). This quantifies how good $\bar{\theta}_{t-1}$ already is as a solution to $F_t$.

In the end, this allows us to use the catalyst general results with primal criterion, and with simple warm-start scheme (warm-start on the last iterate of the last outer iteration). The first point is presented at the beginnning of this section and the second one is adressed by Lemma 5. The following section deals the last point.

## C.2  Proof of Theorem 5

We now show a bound on the initial error of an inner loop when warm-starting on the last iterate of the previous inner loop. Indeed, the convergence results for DVR depend on the initial dual error and so results from [23] cannot be used directly. Yet, it can be adapted, as we show in this section. We note $D_t(\lambda)$ the dual function at outer step $t$ (which should not be mistaken with the Bregman divergence $D_\phi$), and $\lambda_\star^t$ its minimizer. Similarly, we note $\theta_\star^t = \arg\min_\theta F_t(\theta)$, whereas $\theta^\star$ is the global minimizer of $F$. The following theorem ensures convergence of $\bar{\theta}_t$ to the true optimum, given that the subproblems are solved precisely enough.

**Theorem 6.** *[23, Proposition 5]. If $F_k(\bar{\theta}_k) - F_k(\theta_\star^k) \leq \varepsilon_k$ for all $k \leq t$ then*

$$F(\bar{\theta}_t) - F(\theta^\star) \leq \frac{8}{(\sqrt{q} - \rho^{\text{out}})^2}(1 - \rho^{\text{out}})^{t+1}(F(\bar{\theta}_0) - F(\theta^\star)). \qquad (38)$$

Therefore, our goal is to prove that $F_t(\bar{\theta}_{t+1}) - F_t(\theta_\star^t) \leq \varepsilon_t$ for all $t$. The smoothness of $F_t$ ensures that this is achieved if

$$\sum_{i=1}^n \|\theta_{t+1}^{(i)} - \theta_\star^t\|^2 \leq \frac{n}{L}\varepsilon_t. \qquad (39)$$

Yet, using Lemma 5, we know that, since $\theta_{t+1}^{(i)}$ is obtained by applying $K$ steps of DVR to $F_t$ starting from $\lambda_0^t$.

$$\sum_{i=1}^{n} \|\theta_{t+1}^{(i)} - \theta_\star^t\|^2 \leq \frac{(\beta + \sigma_{\max} + L_{\max})}{(\sigma_{\min} + \beta)^2} (1-\rho)^K \left( \frac{p_{\min}}{\eta_t} D_\phi(\lambda_\star^t, \lambda_0^t) + D_t(\lambda_\star^t) - D_t(\lambda_0^t) \right).$$

Unfortunately, we have no control over the dual error at this point. In the remainder of this section, we prove by recursion that Equation (39) holds for all $t$. More specifically, we start by assuming that:

$$\frac{1}{2} \sum_{i=1}^{n} \|\theta_{t+1}^{(i)} - \theta_\star^t\|^2 \leq \frac{n}{L} \varepsilon_t, \tag{40}$$

$$\frac{1}{2} \sum_{i=1}^{n} \sum_{j=1}^{m} \|\theta_{t+1}^{(ij)} - \theta_\star^t\|^2 \leq C_1 \varepsilon_t, \tag{41}$$

$$D_t(\lambda_\star^t) - D_t(\lambda_{t+1}) \leq C_2 \varepsilon_t, \tag{42}$$

where $C_1$ and $C_2$ are such that the conditions are verified for $t = -1$, with $D_{-1} = D_0$, $\theta_\star^{-1} = \theta_\star^0$, and $\lambda_\star^{-1} = \lambda_\star^0$. Equation (40) may not hold for $t = -1$, but making it hold at time $t = 0$ would only require a slightly longer first inner iteration, meaning at most an extra log factor. Therefore we assume without loss of generality that it is the case, since the final complexities are given up to logarithmic factors. The rest of this section is devoted to showing that if $K$ is chosen as in Theorem 5 then Equations (40), (41) and (42) hold regardless of $t$. The first part focuses on assessing the initial error of outer iteration $t + 1$ when the conditions hold at the end of outer iteration $t$, and the second part on showing how these errors shrink during outer iteration $t + 1$.

### C.2.1   Warm-start error

We know that DVR converges linearly, and so the error for each subproblem decreases exponentially fast. Yet, we need to know how big the error is when solving a new problem in order to make sure that the progress from solving previous subproblems is not lost. The point of this is to avoid an extra $\log(\varepsilon^{-1})$ factor in the rate, which would come from having to solve each subproblem from a $O(1)$ precision to an $\varepsilon$ precision using DVR. We show in this section that the initial error is actually much lower than $O(1)$ and decreases with the outer iterations. We first start by bounding the variations of $\omega_t$ across iterations, which we will need for the next proofs.

**Lemma 6** (Distance between subproblems). *It holds that*

$$\|\omega_t - \omega_{t-1}\|^2 \leq C_\omega \varepsilon_{t-1}, \text{ with } C_\omega = \frac{1080n}{1 - \rho^{\text{out}}} \left( \frac{8(1 - \rho^{\text{out}})}{\sigma_{\min}(\sqrt{q} - \rho^{\text{out}})^2} + \frac{4}{9L} \right).$$

*Proof.* The form of the updates yields that (see [23, Proposition 12] or [20, Proof of Lemma 10])

$$\|\omega_t^{(i)} - \omega_{t-1}^{(i)}\| \leq 40 \max\{\|\theta_t^{(i)} - \theta^\star\|, \|\theta_{t-1}^{(i)} - \theta^\star\|, \|\theta_{t-2}^{(i)} - \theta^\star\|\}.$$

Note that here, $\theta^\star$ is the actual solution of the primal problem without the catalyst perturbation. Then, the error can be decomposed as:

$$\sum_{i=1}^{n} \|\theta_t^{(i)} - \theta^\star\|^2 \leq 3 \sum_{i=1}^{n} \left( \|\theta_t^{(i)} - \theta_\star^t\|^2 + \|\theta_\star^t - \bar{\theta}_t\|^2 + \|\bar{\theta}_t - \theta^\star\|^2 \right)$$

$$\leq 3n\|\bar{\theta}_t - \theta^\star\|^2 + 6 \sum_{i=1}^{n} \|\theta_t^{(i)} - \theta_\star^t\|^2.$$

Finally, the strong convexity of $F$ leads to

$$\frac{\sigma_{\min}}{2} \|\bar{\theta}_t - \theta^\star\|^2 \leq F(\bar{\theta}_t) - F(\theta^\star) \leq \frac{8}{(\sqrt{q} - \rho^{\text{out}})^2} (1 - \rho^{\text{out}})^{t+1} (F(\theta_0) - F(\theta^\star)) \tag{43}$$

where in the last inequality we use [23, Proposition 5], which holds because $F_k(\bar{\theta}_k) - F_k(\theta_\star^k) \leq \varepsilon_k$ for all $k < t$. Indeed, $K$ is such that for all $k \leq t$, $\frac{1}{2} \sum_{i=1}^{n} \|\theta_k^{(i)} - \theta_\star^k\|^2 \leq \frac{n}{L} \varepsilon_k$, which yields:

$$F_k(\bar{\theta}_k) - F_k(\theta_\star^k) \leq \frac{L}{2} \|\bar{\theta}_k - \theta_\star^k\|^2 \leq \frac{L}{2n} \sum_{i=1}^{n} \|\theta_k^{(i)} - \theta_\star^k\|^2 \leq \varepsilon_k.$$

Therefore,

$$\sum_{i=1}^{n} \|\theta_t^{(i)} - \theta^\star\|^2 \leq 6n \left(1 - \rho^{\mathrm{out}}\right)^t \left(F(\theta_0) - F(\theta^\star)\right) \left(\frac{8(1 - \rho^{\mathrm{out}})}{\sigma_{\min}(\sqrt{q} - \rho^{\mathrm{out}})^2} + \frac{4}{9L}\right),$$

and a similar bound can be used for $\theta_{t-1}^{(i)}$ and $\theta_{t-2}^{(i)}$. Then, we finish proof by plugging in the expression of $\varepsilon_{t-1}$. $\qquad\square$

We then use Lemma 6 to bound the initial dual error. We denote $\theta_k^t$ (and $\lambda_k^t$) the parameters at inner iteration $k$ of outer iteration $t$.

**Lemma 7** (Dual error warm-start). *The warm-started dual error verifies:*

$$D_t(\lambda_\star^t) - D_t(\lambda_t) \leq C_D \varepsilon_{t-1}, \text{ with } C_D = \left(C_2 + C_\omega + 4\frac{\beta n}{L}\right). \tag{44}$$

Note that we simply warm-start the dual coordinates for an outer iteration using the last iterate from the previous one. Yet, this leads to $\theta_0^t = \theta_K^{t-1} + \beta \Sigma_\beta^{-1}(\omega_{t+1} - \omega_t)$, as in Algorithm 2.

*Proof.* Equation (33) implies that $D_t(\lambda)$ can be written as:

$$D_t(\lambda) = -\sum_{i=1}^{n} \frac{1}{\beta + \sigma_i} \left[\frac{1}{2}(A\lambda)^{(i)} + \beta\omega_t^{(i)}\right]^\top (A\lambda)^{(i)} + R_{\mathrm{comp}}(\lambda), \tag{45}$$

with $R_{\mathrm{comp}}(\lambda)$ that only depends on $\lambda^{(ij)}$ and not on $\omega_t^{(i)}$ for $i \in \{1, \cdots, n\}$. Therefore,

$$D_t(\lambda_\star^t) - D_t(\lambda_K^{t-1})$$

$$= D_{t-1}(\lambda_\star^t) - D_{t-1}(\lambda_K^{t-1}) - \beta \sum_{i=1}^{n} \left[(A\lambda_\star^t)^{(i)} - (A\lambda_K^{t-1})^{(i)}\right]^\top \Sigma_\beta \left[\omega_t^{(i)} - \omega_{t-1}^{(i)}\right].$$

Equation (29) writes $(A\lambda_\star^t)^{(i)} = (\beta + \sigma_i)\theta_\star^t - \beta\omega_t^{(i)}$, and so:

$$A\lambda_\star^t - A\lambda_K^{t-1} = A\lambda_\star^t - A\lambda_\star^{t-1} + A\lambda_\star^{t-1} - A\lambda_K^{t-1} = \Sigma_\beta^{-1}(\theta_\star^t - \theta_\star^{t-1}) + A\lambda_\star^{t-1} - A\lambda_K^{t-1} - \beta(\omega_t - \omega_{t-1}).$$

Then, we know from the equivalent reformulation of Equation (34) that $\theta_t^\star = \arg\min F(\theta) + \frac{\beta}{2}\|\theta - \bar\omega_t\|^2$, so using the 1-Lipschitzness of the proximal operator yields

$$\|\theta_\star^t - \theta_\star^{t-1}\|^2 \leq \|\bar\omega_t - \bar\omega_{t-1}\|^2 \leq \frac{1}{n}\sum_{k=1}^{n} \|\omega_t^{(k)} - \omega_{t-1}^{(k)}\|^2 = \frac{1}{n}\|\omega_t - \omega_{t-1}\|^2. \tag{46}$$

Similarly, $\Sigma_\beta(A\lambda_\star^{t-1} - A\lambda_K^{t-1}) = \theta_\star^{t-1} - (\theta_K^{t-1})^{(i)}$, and so:

$$\sum_{i=1}^{n} \left[(A\lambda_\star^t)^{(i)} - (A\lambda_K^{t-1})^{(i)}\right] \Sigma_\beta \left[\omega_t^{(i)} - \omega_{t-1}^{(i)}\right] \leq \sum_{i=1}^{n} \left\|\frac{(A\lambda_\star^t)^{(i)} - (A\lambda_K^{t-1})^{(i)}}{\beta + \sigma_i}\right\| \left\|\omega_t^{(i)} - \omega_{t-1}^{(i)}\right\|$$

$$\sum_{i=1}^{n} 2\|\theta_\star^t - \theta_\star^{t-1}\|^2 + 2\|\theta_\star^{t-1} - (\theta_K^{t-1})^{(i)}\|^2 + \left(\frac{\beta}{\beta + \sigma_i} + 4\right)\|\omega_t^{(i)} - \omega_{t-1}^{(i)}\|^2.$$

Plugging in Equation (46) yields:

$$D_t(\lambda_\star^t) - D_t(\lambda_K^{t-1}) \leq D_{t-1}(\lambda_\star^t) - D_{t-1}(\lambda_K^{t-1}) + 2\beta \sum_{i=1}^{n} \|(\theta_K^{t-1})^{(i)} - \theta_\star^{t-1}\|^2 + 7\beta\|\omega_t - \omega_{t-1}\|^2$$

Finally note that $D_{t-1}(\lambda_\star^t) \leq D_{t-1}(\lambda_\star^{t-1})$ since $\lambda_\star^{t-1}$ is the maximizer of $D_{t-1}$, and $(\theta_K^{t-1})^{(i)} = \theta_t^{(i)}$ since it is the output of DVR after inner iteration $t$. The final expression is obtained using 6 and the recursion assumptions given by Equations (40) and (42). $\qquad\square$

Finally, the warm-start error on the nodes parameters is given by the two following lemmas.

**Lemma 8** (Virtual parameters warm-starts)**.** *Denote* $\|\theta_1 - \theta_2\|^2_{\text{comp}} = \sum_{i=1}^{n} \sum_{j=1}^{m} \|\theta_1^{(ij)} - \theta_2^{(ij)}\|^2$. *Then,*

$$\|\theta_0^t - \theta_\star^t\|^2_{\text{comp}} \leq 2(C_\omega + 2mC_1)\varepsilon_{t-1}. \tag{47}$$

*Proof.* We use the fact that $(\theta_t)^{(ij)} = (\theta_0^t)^{(ij)} = (\theta_K^{t-1})^{(ij)}$ to write:

$$\|\theta_0^t - \theta_\star^t\|^2_{\text{comp}} = \|\theta_K^{t-1} - \theta_\star^{t-1} + \theta_\star^{t-1} - \theta_\star^t\|^2_{\text{comp}} \leq 2\|\theta_t - \theta_\star^{t-1}\|^2_{\text{comp}} + 2nm\|\theta_\star^{t-1} - \theta_\star^t\|^2.$$

Then, as before, the 1-Lipchitzness of the prox operator yields $\|\theta_\star^{t-1} - \theta_\star^t\| \leq \frac{1}{n}\|\omega_t - \omega_{t-1}\|$. □

**Lemma 9** (Parameters warm-start)**.** *Denote* $\|\theta_1 - \theta_2\|^2_{\text{comp}} = \sum_{i=1}^{n} \sum_{j=1}^{m} \|\theta_1^{(ij)} - \theta_2^{(ij)}\|^2$. *Then,*

$$\sum_{i=1}^{n} \|(\theta_0^t)^{(i)} - \theta_\star^t\|^2 \leq 6\left(C_\omega + \frac{n}{L}\right)\varepsilon_{t-1}. \tag{48}$$

*Proof.* We use the fact that since $\lambda_0^t = \lambda_K^{t-1}$ then $(\theta_0^t)^{(i)} = (\theta_0^t)^{(i)} + \frac{\beta}{\beta + \sigma_i}(\omega_t^{(i)} - \omega_{t-1}^{(i)})$ to write:

$$\sum_{i=1}^{n} \|(\theta_0^t)^{(i)} - \theta_\star^t\|^2 \leq \sum_{i=1}^{n} \|(\theta_K^{t-1})^{(i)} - \theta_\star^{t-1} + \theta_\star^{t-1} - \theta_\star^t + \frac{\beta}{\sigma_i + \beta}(\omega_t^{(i)} - \omega_{t-1}^{(i)})\|^2$$

$$\leq 3\|\omega_t - \omega_{t-1}\|^2 + 3n\|\theta_\star^{t-1} - \theta_\star^t\|^2 + 3\sum_{i=1}^{n} \|(\theta_t)^{(i)} - \theta_\star^{t-1}\|^2.$$

□

We finish this part on warm starts by proving the following lemma, that links the initial dual parameters error (computed with the Bregman divergence of $\phi$), to the other parameters which we already know how to control.

**Lemma 10** (Dual parameters warm-start, as measured by the Bregman divergence)**.**

$$D_\phi(\lambda_\star^t, \lambda_0^t) \leq C_\phi \varepsilon_{t-1}, \tag{49}$$

*with* $C_\phi = \frac{6(C_\omega + n/L) + L_{\max}^2(2C_\omega + 2mC_1)}{\lambda_{\min}^+(A^\top \Sigma_\beta^2 A)} + \frac{2L_{\max}(C_\omega + 2mC_1)}{\alpha}$.

*Proof.* We first decompose the Bregman divergence as:

$$D_\phi(\lambda_0^t, \lambda_\star^t) \leq \frac{1}{2}\|(\lambda_0^t)^{\text{comm}} - (\lambda_\star^t)^{\text{comm}}\|_{A_{\text{comm}}^\dagger A_{\text{comm}}} + \sum_{i=1}^{n} \sum_{j=1}^{m} D_{\phi_{ij}}((\lambda_\star^t)^{(ij)}, (\lambda_0^t)^{(ij)}). \tag{50}$$

Then, we bound the communication term as:

$$\|(\lambda_0^t)^{\text{comm}} - (\lambda_\star^t)^{\text{comm}}\|_{A_{\text{comm}}^\dagger A_{\text{comm}}} \leq \|\lambda_0^t - \lambda_\star^t\|_{A^\dagger A} \leq \frac{1}{\lambda_{\min}^+(A^\top \Sigma_\beta^2 A)}\|\Sigma_\beta A \left(\lambda_0^t - \lambda_\star^t\right)\|^2$$

$$= \frac{1}{\lambda_{\min}^+(A^\top \Sigma_\beta^2 A)}\left(\sum_{i=1}^{n} \|(\theta_0^t)^{(i)} - \theta_\star^t\|^2 + \sum_{i=1}^{n}\sum_{j=1}^{m} \mu_{ij}^2\|(\lambda_0^t)^{(ij)} - (\lambda_\star^t)^{(ij)}\|^2\right)$$

$$= \frac{1}{\lambda_{\min}^+(A^\top \Sigma_\beta^2 A)}\left(\sum_{i=1}^{n} \|(\theta_0^t)^{(i)} - \theta_\star^t\|^2 + \sum_{i=1}^{n}\sum_{j=1}^{m} \|\nabla f_{ij}((\theta_0^t)^{(ij)}) - \nabla f_{ij}((\theta_\star^t)^{(ij)})\|^2\right)$$

$$= \frac{1}{\lambda_{\min}^+(A^\top \Sigma_\beta^2 A)}\left(\sum_{i=1}^{n} \|(\theta_0^t)^{(i)} - \theta_\star^t\|^2 + \sum_{i=1}^{n}\sum_{j=1}^{m} L_{ij}^2\|(\theta_0^t)^{(ij)} - (\theta_\star^t)^{(ij)}\|^2\right).$$

Using Lemmas 8 and 9, we obtain:

$$\frac{1}{2}\|(\lambda_0^t)^{\text{comm}} - (\lambda_\star^t)^{\text{comm}}\|_{A_{\text{comm}}^\dagger A_{\text{comm}}} \leq \frac{\left(6\left(C_\omega + \frac{n}{L}\right) + L_{\max}^2(2C_\omega + 2mC_1)\right)}{\lambda_{\min}^+(A^\top \Sigma_\beta^2 A)}\varepsilon_{t-1}. \tag{51}$$

For the computation part, we use the duality property of the Bregman divergence, which yields

$$
\begin{aligned}
D_{\phi_{ij}}((\lambda_\star^t)^{(ij)}, (\lambda_0^t)^{(ij)}) &= \frac{L_{ij}}{\mu_{ij}^2} D_{f_{ij}^*}(\mu_{ij}(\lambda_\star^t)^{(ij)}, \mu_{ij}(\lambda_0^t)^{(ij)}) \\
&= \frac{L_{ij}}{\mu_{ij}^2} D_{f_{ij}}(\nabla f_{ij}(\mu_{ij}(\lambda_0^t)^{(ij)}), \nabla f_{ij}(\mu_{ij}(\lambda_\star^t)^{(ij)})) \\
&= \frac{L_{ij}}{\mu_{ij}^2} D_{f_{ij}}((\theta_0^t)^{(ij)}, (\theta_\star^t)^{(ij)}) \leq \frac{L_{ij}^2}{\mu_{ij}^2} \|(\theta_0^t)^{(ij)} - (\theta_\star^t)^{(ij)}\|^2
\end{aligned}
$$

Therefore,

$$
\sum_{i=1}^n \sum_{j=1}^m D_{\phi_{ij}}((\lambda_0^t)^{(ij)}, (\lambda_\star^t)^{(ij)}) \leq \frac{2L_{\max}(C_\omega + 2mC_1)}{\alpha} \varepsilon_{t-1}. \tag{52}
$$

Substituting Equations (51) and (52) into Equation (50) finishes the proof. $\qquad\square$

### C.2.2 Inner iteration error decrease

Now that we have bounded the error at the beginning of each outer iteration, we bound error at the end of each outer iteration by using the convergence results for DVR. We first prove the following Lemma, which controls the distance between the virtual parameters and the actual one:

**Lemma 11** (Virtual error decrease). *For all $(i, j)$,*

$$
\mathbb{E}\left[\sum_{i,j} \|(\theta_{t+1})^{(ij)} - \theta_\star^t\|^2\right] \leq (1-\rho)^K \left[\|\theta_0^t - \theta_\star^t\|_{\mathrm{comp}}^2 + \frac{\rho_{\mathrm{sum}} K}{1-\rho} C_0(t)\right]. \tag{53}
$$

*Proof.* We cannot retrieve direct control over the $\theta_{t+1}^{(ij)}$ from control over the dual variables or the dual error, since this would require the $f_{ij}^*$ functions to be smooth, which they may not be. Yet, we leverage the fact that $\theta_{t+1}^{(ij)}$ is obtained by a convex combination between $\theta_t^{(ij)}$ and $\theta_t^{(i)}$ to obtain convergence of to $\theta_\star^t$. We note $j_{t,k}(i)$ the virtual node that is updated at time $(t, k)$ for node $i$. We note $\mathbb{E}_k$ the expectation relative to the value of $j_{t,k}(i)$. We start by remarking that:

$$
\begin{aligned}
&\mathbb{E}_{k+1}\left[\|(\theta_{k+1}^t)^{(ij)} - \theta_\star^t\|^2\right] \\
&= (1-p_{ij})\|(\theta_k^t)^{(ij)} - \theta_\star^t\|^2 + p_{ij}\|(1-\rho_{ij})(\theta_k^t)^{(ij)} + \rho_{ij}(\theta_k^t)^{(i)} - \theta_\star^t\|^2 \\
&\leq (1-p_{ij}\rho_{ij})\|(\theta_k^t)^{(ij)} - \theta_\star^t\|^2 + p_{ij}\rho_{ij}\|(\theta_k^t)^{(i)} - \theta_\star^t\|^2,
\end{aligned}
$$

where in the last inequality we used the convexity of the squared norm. We use that $p_{ij}\rho_{ij} \geq \rho$ (equal for the smallest one), and write that:

$$
\mathbb{E}\left[\|(\theta_K^t)^{(ij)} - \theta_\star^t\|^2\right] \leq (1-\rho)^K \|(\theta_0^t)^{(ij)} - \theta_\star^t\|^2 + p_{ij}\rho_{ij} \sum_{k=1}^K (1-\rho)^{k-1}\|(\theta_{K-k}^t)^{(i)} - \theta_\star^t\|^2. \tag{54}
$$

Noting $\rho_{\mathrm{sum}} = \max_i \sum_{j=1}^m \rho_{ij}p_{ij}$ and $\|\theta_k^t - \theta_\star^t\|_{\mathrm{comp},i}^2 = \sum_{j=1}^m \|(\theta_k^t)^{(ij)} - \theta_\star^t\|^2$, we obtain

$$
\mathbb{E}\left[\|\theta_K^t - \theta_\star^t\|_{\mathrm{comp},i}^2\right] \leq (1-\rho)^K \|\theta_0^t - \theta_\star^t\|_{\mathrm{comp},i}^2 + \rho_{\mathrm{sum}} \sum_{k=1}^K (1-\rho)^{k-1}\|(\theta_{K-k}^t)^{(i)} - \theta_\star^t\|^2. \tag{55}
$$

Using Lemma 5, we know that $\sum_{i=1}^n \|(\theta_k^t)^{(i)} - \theta_\star^t\|^2 \leq C_0(t)(1-\rho)^k$, with $C_0(t)$ a constant that depends on the initial conditions of outer iteration $t$. Therefore,

$$
\sum_{i=1}^n \sum_{k=1}^K (1-\rho)^{k-1}\|(\theta_{K-k}^t)^{(i)} - \theta_\star^t\|^2 \leq K(1-\rho)^{K-1}C_0(t). \tag{56}
$$

In the end,

$$
\mathbb{E}\left[\|\theta_K^t - \theta_\star^t\|_{\mathrm{comp}}^2\right] \leq (1-\rho)^K \left[\|\theta_0^t - \theta_\star^t\|_{\mathrm{comp}}^2 + \frac{\rho_{\mathrm{sum}} K}{1-\rho} C_0(t)\right]. \tag{57}
$$

$\qquad\square$

This lemma has the following corollary:

**Corollary 1** (Warm-started virtual error decrease). *For all $(i, j)$,*

$$\mathbb{E}\left[\sum_{i,j} \|(\theta_{t+1})^{(ij)} - \theta_\star^t\|^2\right] \leq (1-\rho)^K \left[6\left(C_\omega + \frac{n}{L}\right) + K\frac{\rho_{\text{sum}}C_{\text{comp}}}{1-\rho}\right]\varepsilon_{t-1}, \quad (58)$$

*with*

$$C_{\text{comp}} = \frac{(\beta + \sigma_{\max} + L_{\max})}{(\sigma_{\min} + \beta)^2}\left(\frac{p_{\min}}{\eta_t}C_\phi + C_2 + C_\omega + 4\frac{\beta n}{L}\right)$$

*Proof.* Using Lemmas 5, 10 and 7, we write:

$$C_0(t) = \frac{(\beta + \sigma_{\max} + L_{\max})}{(\sigma_{\min} + \beta)^2}\left(\frac{p_{\min}}{\eta_t}D_\phi(\lambda_\star^t, \lambda_0^t) + \left(D(\lambda_\star^t) - D(\lambda_0^t)\right)\right) \leq C_{\text{comp}}\varepsilon_{t-1}$$

We use Lemma 8 for the first term. □

**Lemma 12** (Condition on $K$). *If Equations* (40), (41) *and* (42) *hold at time $t$, and $K$ is such that:*

$$(1-\rho)^K \leq \min\left(\frac{C_1(1-\rho^{\text{out}})}{12(C_\omega + n/L)}, \frac{C_1(1-\rho^{\text{out}})(1-\rho)}{K\rho_{\text{sum}}C_{\text{comp}}}, \frac{C_2}{C_L}, \frac{n(\sigma_{\min} + \beta)^2}{2LC_L(\beta + \sigma_{\max} + L_{\max})}\right),$$

*then they also hold at time $t + 1$.*

*Proof.* Using Corollary 1, we obtain that if $K$ is set such that

$$(1-\rho)^K\left[6\left(C_\omega + \frac{n}{L}\right) + K\frac{\rho_{\text{sum}}C_{\text{comp}}}{1-\rho}\right] \leq C_1(1-\rho^{\text{out}}),$$

then the recursion condition is respected for the virtual parameters. This yields the first and second conditions on $K$. Now, we write $C_L = \left(\frac{p_{\min}}{\eta_t}C_\phi + C_D\right)$, then using Lemmas 10 and 7 (where $C_\phi$ and $C_D$ are defined), we obtain using Theorem 4 that

$$D_t(\lambda_\star^t) - D_t(\lambda_{t+1}) \leq C_L(1-\rho)^K\varepsilon_{t-1},$$

since $\lambda_{t+1}$ is obtained by performing $K$ iterations of DVR to minimize $F_t$ starting from $\lambda_t$. This yields the third condition on $K$. Finally, the last condition on $K$ is obtained by leveraging Lemma 5.

□

# D Experiments

For the experiments, the following logistic regression problem is solved:

$$\min_{\theta \in \mathbb{R}^d} \sum_{i=1}^n \left[\frac{\sigma}{2}\|\theta\|^2 + \sum_{j=1}^m \frac{1}{m}\log(1 + \exp(-y_{ij}X_{ij}^\top\theta))\right], \quad (59)$$

where the pairs $(X_{ij}, y_{ij}) \in \mathbb{R}^d \times \{-1, 1\}$ are taken from the RCV1 dataset, which we downloaded from https://www.csie.ntu.edu.tw/~cjlin/libsvmtools/datasets/binary.html.

Figure 3 is the full version of Figure 1, in which we report the number of individual gradients and number of communications for each configuration. We see that accelerated EXTRA actually outperforms EXTRA when the regularization is small, as already mentioned in the main text. We also see that Accelerated EXTRA and Accelerated DVR have comparable communication complexity on the grid graph, when $\gamma$ is smaller. Yet, the computation complexity of (accelerated) DVR is much smaller, so accelerated DVR is much faster overall as long as $\tau$ is not too big.

(a) Erdős-Rényi, $\sigma = m \cdot 10^{-5}$

(b) Grid, $\sigma = m \cdot 10^{-5}$

(c) Erdős-Rényi, $\sigma = m \cdot 10^{-7}$

(d) Grid, $\sigma = m \cdot 10^{-7}$

Figure 3: Experimental results for the RCV1 dataset with different graphs of size $n = 81$, with $m = 2430$ samples per node, and with different regularization parameters.