[Reviews · NeurIPS 2020]

Review 1

Summary and Contributions: The authors developed a stochastic decentralized algorithm whose performance is better than the state of the art algorithms such as EXTRA. In general, first-order methods can be improved using duality however, solving the subproblem also requires extra computation and communication (also mentioned by the authors). In this paper, the authors use Bergman divergence in a smart way to avoid these costs coming from dual formulation and at the same time achieving the performance, i.e. convergence, improvements at the algorithm.

Strengths: I believe the novelty of this paper is the use of Bergman divergence to make algorithm dual free (i.e. dual free trick). This algorithm with acceleration provides a competitive algorithm with a fast convergence rate and can be used by the Neurips community.

Weaknesses: I couldn't think of any limitations for the algorithm.

Correctness: I have one question regarding the result of Theorem 1. I may have mistaken but sampling with probability p_{ij} and updating theta make theta random? If I am not wrong then the result of Theorem 1 should be on the expectation of norm of theta.

Clarity: Yes, the paper is well written. I was able to clearly follow the flow of the paper and the proofs.

Relation to Prior Work: Yes. Authors provided sufficient information about the prior work done in the field and the difference of their method.

Reproducibility: Yes

Additional Feedback: - It would be interesting to see the comparison between accelerated DVR and multi-step dual acceleration method, which is introduced by Scaman et al. and also achieves the optimal convergence rate in decentralized setup.


Review 2

Summary and Contributions: This paper introduces a dual-free Decentralized stochastic algorithm with Variance Reduction (DVR), based on a careful choice of Bregman divergence. An accelerated version of DVR is also proposed by adapting the technique of Catalyst and Chebyshev polynomials. Experiments on regularized logistic regression seem to show that DVR and its accelerated variants outperform other baselines.

Strengths: 1. A dual-free decentralized stochastic algorithm with Variance Reduction and it accelerated variant. Significance: Medium. 2. Experiments on regularized logistic regression. Significance: High.

Weaknesses: 1. It is not immediately clear to me what is the main technical challenge to extend the dual-free analysis in [33] and [16] into the decentralized setting. The authors may need to explain this clearly. 2. When is it the case that the evaluating the gradient of conjugate function is expensive or infeasible? It is better to include more examples in machine learning to motivate this dual-free approach.

Correctness: To the best of my knowledge, the theoretical results are correct. The methodology of empirical studies is also sound.

Clarity: Yes, it is easy to read for most of the parts. My suggestion is to highlight the new technical contribution of this work compared with other previous works.

Relation to Prior Work: It discussed prior work extensively, but it is not immediately clear to me what is the main technical contribution compared with previous works.

Reproducibility: Yes

Additional Feedback:


Review 3

Summary and Contributions: This paper studies regularized empirical risk minimization in a decentralized setting over a network of machines and proposes a decentralized stochastic algorithm with variance reduction dubbed as DVR. The authors show that by only computing local stochastic gradients, a linear speedup (n times faster where n is the number of machines) can be achieved. The main idea is to reduce the problem into a dual-free algorithm by utilizing a recently proposed Bregman coordinate descent on a specific dual problem.

Strengths: While the main pillars of the proposed ideas, i.e., dual free and variance reduction are already known, but I think this paper provides nice theoretical results with corroborating empirical studies that fill in a gap in decentralized optimization.

Weaknesses: NA

Correctness: The paper is technically sound and the proofs seem to be correct as far as I checked.

Clarity: The presentation of the paper was mostly clear. The paper is reasonably well written, the structure and language are good and overall the paper is an easy and enjoyable reading.

Relation to Prior Work: The claimed contributions are discussed in the light of existing results and the paper does survey related work appropriately.

Reproducibility: Yes

Additional Feedback:

[Author Response · NeurIPS 2020]

We would like to thank the reviewers for their positive and helpful feedback. We adress the main questions below.

**Typo:** The result of Theorem 1 is on the expected norm of $\theta$ indeed, thank you for pointing this out.

**Technical Challenge for the dual-free analysis**: Our algorithm is inspired from [33] in the sense that we choose
specific Bregman divergences (with respect to the $f_{ij}^*$) in order to obtain closed-form solutions of dual Bregman gradient
steps as gradients of the primal functions $\nabla f_{ij}$. Yet, [33] applies a randomized primal-dual algorithm with fixed
Bregman divergences choice to a specific *primal-dual* formulation. Instead, we apply a generic randomized Bregman
coordinate descent algorithm to a specific *dual* formulation (which has an augmented graph interpretation as explained
in [11]). The main technical challenges that we faced were:

• Computing the relative strong convexity / smoothness constants, which now depend on the topology of the
graph since our decentralized dual formulation is more complex than the formulation from [33].

• Proving convergence of Bregman Coordinate Descent in the relatively smooth setting. Although a similar
algorithm is analyzed in Hanzely and Richtárik [2018], we give sharper results in the case of arbitrary sampling
of blocks, and tightly adapt to the separability structure. This is crucial to our analysis since the probabilities to
sample a local gradient and to communicate can be vastly different. Derivations can be found in Appendix A.

• Proving Catalyst acceleration: Theorem 4 (Appendix B) controls dual variables but we apply Catalyst to the
primal variables, which thus requires both primal and dual warm-start errors, which is done in Appendix C.

Although dual-free SDCA is a dual approach in the spirit, the analysis in [33] is primal, which in particular explains
why individual convexity is not needed. Yet, and although it leads to a related algorithm, our approach is different.

**When is $\nabla f^*$ expensive to compute?** For $x \in \mathbb{R}^d$, $\nabla f^*(x) = \arg\max_y x^\top y - f^*(y)$ so in general, computing the
gradient of the conjugate is as hard as minimizing the function itself. Solving this subproblem requires inverting a $d \times d$
matrix for ridge regression, and it has no closed form solution for logistic regression. Although the dimensionality
of the subproblems can be reduced in the case of stochastic algorithms for linear models, this leads to more complex
implementations and increases numerical errors. In the MSDA implementation below, we obtained the dual gradients
by solving each local subproblem up to precision $10^{-11}$ using accelerated gradient descent. Solving the subproblems
with lower precision caused MSDA to plateau and not converge to the true optimum.

Figure 1: Experimental results for the RCV1 dataset with different graphs of size $n = 81$, with $m = 2430$ samples per node, and with different regularization parameters.

**Comparison with MSDA**: Figure 1 presents the comparison between DVR and MSDA in terms of communication
complexity. In Figure 1(c), *Acc. DVR, comm* (the brown line) refers to Accelerated DVR with Catalyst parameter
chosen to favor communication complexity (as explained after Theorem 2). MSDA is the fastest algorithm as expected,
but accelerated DVR is not too far behind, especially given the fact that it relies on generic Catalyst acceleration, which
adds some complexity overhead. Therefore, the comparison with MSDA corroborates the fact that accelerated DVR is
competitive with optimal methods in terms of communication while enjoying a drastically lower computational cost.

We will make sure to insist on the points discussed in this rebuttal in a revised version of the paper.

# References

F. Hanzely and P. Richtárik. Fastest rates for stochastic mirror descent methods. *arXiv:1803.07374*, 2018.


[Meta-Review · NeurIPS 2020]

The paper integrates some existing techniques, such as dual free trick and variance reduction technique. The novelty at the theoretically side is OK but not very high. The experiments validate the effectiveness of the proposed algorithm. It received three positive scores: 8,6,7, with relatively high confidence. The AC thus deemed that the paper can be accepted.